# Human presence and infrastructure impact wildlife nocturnality differently across an assemblage of mammalian species

**Michael Procko**[1]*, **Robin Naidoo**[2,3], **Valerie LeMay**[1], **A. Cole Burton**[1]

**1** Department of Forest Resources Management, University of British Columbia, Vancouver, British Columbia, Canada, **2** World Wildlife Fund - US, Washington, District of Columbia, United States of America, **3** Institute for Resources, Environment and Sustainability, University of British Columbia, Vancouver, British Columbia, Canada

* xprockox@gmail.com

**Data Availability Statement:** All data are available at Dryad (https://doi.org/10.5061/dryad. w3r2280v7).

## Abstract

Wildlife species may shift towards more nocturnal behavior in areas of higher human influence, but it is unclear how consistent this shift might be. We investigated how humans impact large mammal diel activities in a heavily recreated protected area and an adjacent university-managed forest in southwest British Columbia, Canada. We used camera trap detections of humans and wildlife, along with data on land-use infrastructure (e.g., recreation trails and restricted-access roads), in Bayesian regression models to investigate impacts of human disturbance on wildlife nocturnality. We found moderate evidence that black bears (*Ursus americanus*) were more nocturnal in response to human detections (mean posterior estimate = 0.35, 90% credible interval = 0.04 to 0.65), but no other clear relationships between wildlife nocturnality and human detections. However, we found evidence that coyotes (*Canis latrans*) (estimates = 0.81, 95% CI = 0.46 to 1.17) were more nocturnal and snowshoe hares (*Lepus americanus*) (estimate = -0.87, 95% CI = -1.29 to -0.46) were less nocturnal in areas of higher trail density. We also found that coyotes (estimate = -0.87, 95% CI = -1.29 to -0.46) and cougars (*Puma concolor*) (estimate = -1.14, 90% CI = -2.16 to -0.12) were less nocturnal in areas of greater road density. Furthermore, coyotes, black-tailed deer (*Odocoileus hemionus*), and snowshoe hares were moderately more nocturnal in areas near urban-wildland boundaries (estimates and 90% CIs: coyote = -0.29, -0.55 to -0.04, black-tailed deer = -0.25, -0.45 to -0.04, snowshoe hare = -0.24, -0.46 to -0.01). Our findings imply anthropogenic landscape features may influence medium to large-sized mammal diel activities more than direct human presence. While increased nocturnality may be a promising mechanism for human-wildlife coexistence, shifts in temporal activity can also have negative repercussions for wildlife, warranting further research into the causes and consequences of wildlife responses to increasingly human-dominated landscapes.

**Funding:** "This research was supported by the British Columbia Ministry of Environment and Climate Change Strategy (BC Parks) under agreements TP20JHQ024 and TP21JHQ013. Additional funding was received through grants to ACB from the Natural Sciences and Engineering Council of Canada and the Canada Research Chairs program, as well as from the University of British Columbia's Faculty of Forestry. The funders had no role in study design, data collection and analysis, decision to publish, or preparation of the manuscript."

## Introduction

### Wildlife behavior and protected areas

The perceived risk of predation is well known to influence wildlife behavior [1, 2], creating a "landscape of fear" where species alter their behavior in, or avoid, regions of higher perceived risk [3, 4]. Human disturbance can also induce fear responses in wildlife, altering feeding times, increasing vigilance or flight responses, or prompting species to select habitats with lower perceived human influence [5–9]. These responses may impact predator-prey dynamics, resulting in shifts in trophic structure due to differential effects on species [10]. Human disturbance may thus play a crucial role in shaping both wildlife behavior and predator-prey dynamics [7–9, 11]. Protected areas (PAs), and specifically PAs created for the purpose of wildlife conservation, are often thought to provide refuge from anthropogenic pressures, lessening negative human impacts on wildlife [12, 13]. However, PAs may be established for a number of reasons including the maintenance of accessible outdoor spaces for the purposes of recreational activity to promote human health [14], the preservation of culturally important landmarks or features [15], or the conservation of unique ecosystems or species (e.g., Garry oak ecosystems in the Gulf Islands National Park Reserve in British Columbia (BC), Canada [16]). Many PAs established and maintained with a mandate to conserve wildlife are also intended to promote recreational activity. Yet, high human visitation rates—a growing feature of many of the world's PAs [17]—may hinder conservation efforts, displacing wildlife from regions intended to safeguard them [13, 18]. Understanding whether and how species can coexist with mounting human pressures, including expansions in nature-based tourism, is fundamental to wildlife management in the Anthropocene [19]. Recreational activity in particular can result in effective habitat loss for sensitive species [20, 21], although more adaptable species may persist in areas of higher visitation by altering their timing of activity to minimize conflicts with people [9, 22, 23]. Furthermore, recreational infrastructure (e.g., trail or road density) may also play a role in shaping wildlife activity patterns [24, 25], underscoring the need to disentangle impacts of both direct human presence and human-related landscape features (e.g., "footprint") on wildlife in PAs [26].

### Nocturnality as a coexistence mechanism

Many species have become increasingly nocturnal in response to human activity [27], thereby implying a potential mechanism by which wildlife are able to coexist with humans through fine-scale temporal segregation [9, 22, 28, 29]. This adaptation may be flexible, with some species shifting to be more nocturnal during periods of time when human activity is higher, while others may be more nocturnal in regions associated with greater levels of human activity (e.g., developed areas [28]). Wildlife responses to human disturbance may also vary with the temporal scale of human activity, with more behaviorally plastic species responding to fine-scale shifts in human presence (e.g., hourly or daily traffic [28, 30]), while other species may exhibit patterns which change with weekly or seasonal trends in anthropogenic pressures [31–33]. However, it remains unclear whether shifts towards nocturnality promote coexistence, as such shifts may involve costs to wildlife, formally referred to as "risk effects" [2]. Risk effects may include reduced reproductive output [34, 35], increased temporal overlap between predator and prey species (causing greater predation risk [36]), increased temporal overlap between competing predators [37], or effective habitat loss for prey unable to navigate compounding pressures from primarily diurnal humans and nocturnal predators [33]. These negative repercussions reinforce the importance of understanding wildlife diel activity shifts in response to anthropogenic pressures, which can further inform wildlife management and conservation.

The ability to become more nocturnal in response to human disturbance varies by species. Some species—especially those with a history of conflict with humans—may be more adaptable [7, 28, 38], while others may predominantly exhibit nocturnal behavior regardless of external influences (e.g., snowshoe hares, *Lepus americanus* [39]). Life history traits may also influence species' responses to human disturbance [26], accentuating the importance of species-specific approaches to investigating wildlife diel activity shifts in response to human activity. Wildlife diel activity patterns can also vary with environmental factors such as lunar phase and forest cover due to associated variation in predation risk or hunting success [1, 39, 40]. Hence, it is crucial to consider a number of factors when investigating anthropogenic effects on wildlife nocturnality. Likewise, reliable methods for investigating these phenomena require careful consideration not only of species-specific traits, but also how to most effectively characterize metrics of both wildlife and human activity.

## Quantifying nocturnality and human disturbance

Wildlife nocturnality can be measured in many ways, for instance as a categorical description of the times of day when animals are active (e.g., day vs. night [41]), as ratios of how often animals are detected during the day vs. night [27], or as continuous measures of times of the day wildlife are active [28], with the latter revealing trends in crepuscular activity which might be lost when using coarse categorical descriptions. However, continuous data on wildlife activities are sometimes difficult to collect. Moreover, human disturbance can be measured through direct observation as a continuous variable (e.g., with trail counters, social media data scrapers, or camera traps (CTs) [42–46]), categorized spatially as areas with high vs. low human use [27], inferred from features of a landscape which facilitate greater human presence such as human footprint or recreation trail density [24, 26, 28], and in some cases, human activity may be simulated experimentally (e.g., using audio playback [8, 9]). Recent reviews have pointed to the utility of continuous measures of human activity to evaluate the shape and magnitude of impacts of outdoor recreation on wildlife [47], and thereby guide recreation management [48]. CTs offer a useful approach to attaining continuous measures of both human (e.g., outdoor recreation) and wildlife activities simultaneously [43–45].

## Objectives and hypotheses

Here, we used CTs deployed throughout a heavily recreated PA and an adjacent university-managed research forest to quantify both direct human presence and wildlife diel activities. We used these data in regression models to investigate whether human activity impacts wildlife nocturnality, testing the hypothesis that all wildlife species seek to minimize interactions with humans by exhibiting increased nocturnality in areas where human activity is greater [27]. Additionally, we also modeled information regarding landscape features such as recreation trails, roads, and proximity to the urban-wildland boundary to test the hypothesis that land-use infrastructure (hereafter "infrastructure") and human presence impact wildlife nocturnality differently [26, 28]. We focused our analyses on six species: black bear (*Ursus americanus*); cougar (*Puma concolor*); black-tailed deer (*Odocoileus hemionus*); snowshoe hare; coyote (*Canis latrans*); and bobcat (*Lynx rufus*). We chose these species as they were known to commonly inhabit our study area [44], and could therefore provide a sufficient number of detections to model. Moreover, these species span a diversity of traits such as body size (large and medium-bodied) and trophic level (predators, mesopredators, and prey), which could reveal species-dependent responses to recreation [26].

We predicted all species except snowshoe hares would be more nocturnal in areas of greater human detections, with hares being mostly nocturnal regardless of external factors [39]. Prior

research in different contexts has concluded that cougars may show immediate temporal avoidance of people [28, 49], and both coyotes and bobcats may shift from primarily crepuscular to nocturnal when humans are present [25, 28, 36]. Likewise, we predicted black bears would segregate temporally from humans due to associated risk of injury or death when selecting for areas of higher human presence [50]. Black-tailed deer are commonly crepuscular, but are known to exhibit little-to-no crepuscular or nocturnal activity in a predator-free area [51]. They have also been noted as being more active at night in areas with recreation when compared to areas with no recreation [25]. Therefore, we predicted that deer might perceive humans similar to non-human predators and consequently shift their diel activity patterns towards dawn or dusk to avoid primarily diurnal humans. We also predicted that human presence would impact cougar, black bear, bobcat, and coyote diel activity patterns to a greater degree than infrastructure, as previous works have shown this pattern for large predators in other systems [28]. We did not have explicit predictions for how black-tailed deer might differentially respond to human presence and infrastructure due to a paucity of literature, but we expected snowshoe hares to lack a response to both human presence and infrastructure due to their aforementioned tendency towards nocturnal behavior regardless of external factors [39].

## Materials and methods

### Ethics statement

Written consent was obtained for all research undertaken, provided under protocol A18-0234 from the University of British Columbia's Animal Care Committee and under protocol H21-01424 from the University of British Columbia's Behavioural Research Ethics Board.

### Study area and sampling design

From March 2019 to September 2020, we collected data from 58 CTs deployed throughout the adjacent landscapes of Golden Ears Provincial Park, BC, Canada (hereafter, "Golden Ears") and the University of British Columbia Malcolm Knapp Research Forest, BC (hereafter, "Malcolm Knapp") (Fig 1). These two areas comprise a temperate rain forest with steep, mountainous terrain, nestled between two large lakes (Pitt Lake, Alouette Lake), which render the park and research forest a unique transition zone between the deep backcountry wilderness and the urban-rural gradient of southern BC. Accordingly, the landscape hosts both young and old-growth canopies consisting largely of western hemlock (*Tsuga heterophylla*), yellow cedar (*Chamaecyparis nootkatensis*), western red cedar (*Thuja plicata*), and Coastal Douglas-fir (*Pseudotsuga menziesii* var. *menziewii*), which tend to grow above understories of vine maple (*Acer circinatum*) and salmonberry (*Rubus spectaclus*). Furthermore, as a protected area and university-managed research forest sitting "on the doorstep" of Vancouver, BC—Canada's third most populous city—Golden Ears and Malcolm Knapp experience a range of human activities. As a 625 km$^2$ park dedicated to the preservation of the natural environment for the enjoyment and inspiration of the general public, Golden Ears hosts high levels of recreational activities, including hiking, camping, mountain biking, horseback riding, fishing, and more, though motorized recreation and hunting are not permitted in the park [52]. Conversely, Malcolm Knapp is a 52 km$^2$ research forest dedicated primarily to the provision of research and educational opportunities for university students and faculty, with mild levels of recreation taking place in more accessible (southern) regions of the forest, as well as forest harvest operations which facilitate the principal functions of education and research. Therefore, the landscape offers an interesting mosaic of varying levels and types of human activity in which to explore the impacts of anthropogenic disturbance on wildlife movement and behavior.

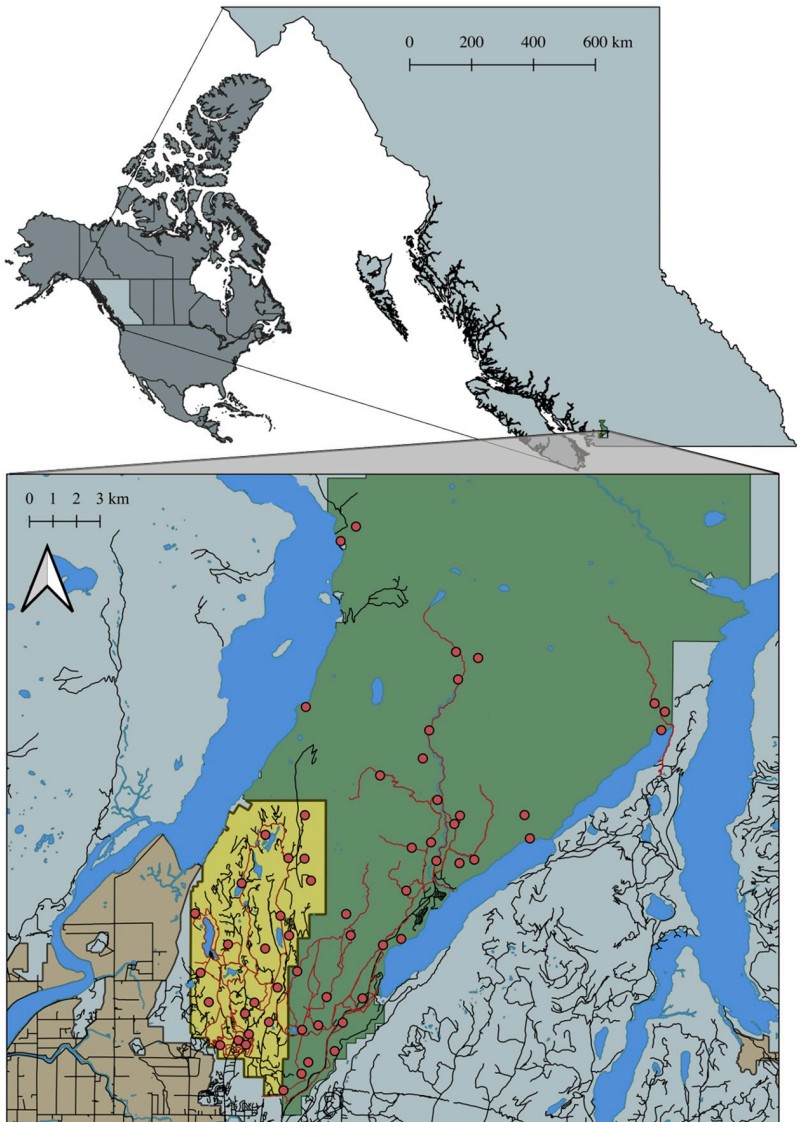

**Fig 1. Study area of Golden Ears Provincial Park (green polygon) and Malcolm Knapp Research Forest (yellow polygon) in southwestern British Columbia, Canada.** Red circles represent camera-trap locations, red lines represent recreation trails, black lines represent roads, blue polygons represent lakes, ponds, and rivers, and brown polygons represent agricultural land reserves. The polygon representing the continent of North America was sourced from the US Department of State, Office of the Geographer (https://geodata.lib.utexas.edu/catalog/stanford-cq068zf3261), and those for all Canadian provinces were sourced from the Government of Canada Open Government Database (https://open.canada.ca/data/en/dataset/a883eb14-0c0e-45c4-b8c4-b54c4a819edb). Polygons representing bodies of water were sourced from the British Columbia Freshwater Atlas (https://catalogue.data.gov.bc.ca/dataset/freshwater-atlas-watersheds), and polygons representing agricultural land reserves were sourced from the British Columbia Provincial Agricultural Land Commission (https://catalogue.data.gov.bc.ca/dataset/alc-agricultural-land-reserve-lines). Polygons representing roads were republished from the British Columbia Digital Road Atlas (https://catalogue.data.gov.bc.ca/dataset/digital-road-atlas-dra-demographic-partially-attributed-roads) under a CC BY license, with permission from the British Columbia Ministry of Environment, original copyright 2023. Polygons representing recreational trails were provided by the British Columbia Ministry of Environment and Climate Change Strategy (BC Parks) internal database.

Our 58 CTs were deployed in a stratified random configuration, with 33 cameras deployed along either recreation trails or restricted-access roads (20 in Golden Ears, 13 in Malcolm Knapp) and 25 cameras deployed "off-trail" along a game trail between 250 m—1 km from both recreation trails and roads (17 in Golden Ears, 8 in Malcolm Knapp). For our off-trail

locations, we used a buffer of 250 m to 1 km due to difficulty accessing locations further off-trail in the steep terrain, thus our inferences do not extend to the more remote regions of the landscape. At each location, we attached a Reconyx Hyperfire Pro 2 (Reconyx, Holmen, WI, USA) to a nearby tree, approximately 3–5 m from the center of the nearest recreation trail, road, or game-trail. We set CTs at approximately 1 m in height to minimize the amount of identifiable human features captured on camera, while maximizing the detection probability of medium-to-large-bodied mammal species. CTs were set to take one photo per trigger, with no delay period, and sensitivity set to "high". We ensured differences between CT settings, heights, and distances to target zones were minimal throughout the study, to alleviate any issues that may arise from differences in detectability caused by these factors [45].

Following our final camera data collections in September 2020, we used a freely-available object detection software, MegaDetector, to categorize our photos into "human", "animal", "vehicle", and "blank" categories [53], which we used to sort our images for more rapid manual classification on a private cloud-based photo management software (WildCo CamTrap System v. 3.0) [54]. We then manually inspected all images, classifying "animal" images by species and ensuring previously categorized "human", "vehicle", and "blank" images were correctly classified by the object detection software. We then imported image detection data to R statistical software (v 3.6.2) for data management and statistical analyses [55].

## Characterizing wildlife nocturnality and human disturbance

To test our hypotheses, we created regression models which contrasted nocturnality against several measures of human disturbance, including direct human presence and infrastructure (model specifics below in "Modeling framework"). We quantified our response variable—nocturnality—from independent detection "events" of wildlife, which we defined as CT images taken at the same camera more than 30 minutes after another image of the same species [56]. For each detection event, we calculated the degree of nocturnality as the absolute value of the time (in decimal hours) between the detection event (using the timestamp of the first image of the event) and solar noon, where solar noon is the time of day the sun appears at its apex in the sky (e.g., nocturnality of 1.50 decimal hours = 1 hour and 30 minutes either before or after solar noon). Given the anticipated greater concentration of human activity during mid-day hours, the response variable therefore represents shifts in wildlife diel activity away from times of peak human use, and prior studies have characterized nocturnality using similar methods [28]. We performed nocturnality calculations using the R package *solartime* (v 0.0.2) [57].

We derived measures of direct human presence from CTs by considering independent detection events of humans as images at the same camera station which were taken more than 1 minute after another image [44], since many recreationists are likely to traverse high-traffic hiking trails within the 30-minute period that we used to define independence between consecutive wildlife detections. We speculated that direct human presence might impact species' behavior at different temporal scales, with some species exhibiting more plastic behavior (e.g., immediate responses to daily human use), while others might exhibit behavioral patterns that are aligned with longer periods of time (e.g., more consistent responses to anticipated seasonal changes in human use), and thus calculated three temporal scales of direct human presence: 1) the number of humans detected at a camera station during the same day as the independent wildlife detection event (hereafter "daily human detections"); 2) the number of human detections at a camera station per week divided by number of days the camera was active that week (hereafter "weekly human detection rate"); and 3) the number of human detections at a camera station per month divided by number of days the camera was active that month (hereafter "monthly human detection rate"). We derived measures of infrastructure from geographic

**Table 1. Predictor variables considered in construction of regression models which modelled the nocturnality of each independent detection event of a given species as a function of a suite of predictor variables.** Prior to modeling, predictor variables were tested for excessive multicollinearity. Due to anticipated excessive multicollinearity (Pearson's r > 0.7) of all direct human presence variables, only one variable from this category was included in each species' model. Direct human presence variable selection was performed by regressing nocturnality against each direct human presence predictor independently, and comparing Bayes Factors of these models against a null (intercept-only) model. The variable from the model with the highest Bayes Factor was selected, so long as Bayes Factor was > 1. If no Bayes Factors were > 1, a direct human presence variable was not included in the species' final model. [a] CT: camera trap, [b] GIS: Geographic Information Systems (Acquisition through ArcGIS Pro), [c] BC Vegetation Resources Inventory (2019 data; (https://www2.gov.bc.ca/gov/content/industry/forestry/managing-our-forest-resources/forest-inventory/data-management-and-access), accessed May 29, 2020.

| Variable | Category | Acquisition | Calculation |
|---|---|---|---|
| Daily human detections | Direct human presence | CT [a] | The number of human detections at that camera during that day |
| Weekly human detection rate | Direct human presence | CT | The total number of human detections at that camera during that week, divided by the number of days that camera was operating that week (accounts for sampling effort) |
| Monthly human detection rate | Direct human presence | CT | The total number of human detections at that camera during that month, divided by the number of days that camera was operating that month (accounts for sampling effort) |
| Dist. to boundary | Infrastructure | GIS [b] | Distance from the southern urban-wildland boundary of the study area (m) |
| Recreation trail density | Infrastructure | GIS | Recreation trail length within a 500 m buffer of the camera station divided by the area of the 500 m buffer $(m/m^2)$ |
| Road density | Infrastructure | GIS | Road length within a 500 m buffer of the camera station divided by the area of the 500 m buffer $(m/m^2)$ |
| Lunar cycle | Environmental | CT | Lunar fraction as a percent (0–1), calculated from timestamp with R package *suncalc* (v 0.5.1) [58] |
| Crown closure | Environmental | GIS | Overlaid BC Vegetation Resources Inventory (VRI[c]) forest cover polygons onto camera station points and extracted the tree crown closure percent (attribute label: CR_CLOSURE) for the point |

information systems (GIS) data (Table 1). These included the distance (m) of the camera station to the nearest southern boundary of either Golden Ears or Malcolm Knapp (i.e., the urban-wildland boundary), and recreation trail density $(m/m^2)$ and road density $(m/m^2)$, both in a 500 m buffer around each camera. We used a 500 m buffer as it represented a large enough area to characterize linear features which might influence animal movement near the camera station, but was small enough to minimize inclusion of recreation trails and roads inaccessible from the location (i.e., across steep cliff faces or wide rivers).

## Modeling framework

Since daily human detections, weekly human detection rates, and monthly human detection rates were highly collinear, and could therefore not be modeled together (S1 Fig), we performed a preliminary "scale analysis" to determine which direct human presence variable best explained variation in nocturnality data. This included constructing three Bayesian linear models for each species to contrast nocturnality responses independently against daily human detections, weekly human detection rates, and monthly human detection rates. We compared these preliminary models against a null, intercept-only model using Bayes Factor to determine the temporal scale at which human activity best explained variation in nocturnality responses, if any. Bayes Factor represents a ratio of the likelihood of a hypothesis against the likelihood of a competing hypothesis [59]—in our case the likelihood of a direct human presence variable explaining more variation in the data than the null model. Therefore, the predictor variable from the preliminary model with the highest Bayes Factor was Included in subsequent models [60], but only if the Bayes Factor was greater than 1. Bayes Factors were calculated using the R package *bayestestR* (v 0.13.0) [61].

We then constructed one final Bayesian linear model for each of our six focal species, with nocturnality as a continuous response variable contrasted against the top-performing direct human presence variable, the three measures of infrastructure, plus environmental variables hypothesized *a priori* as having a possible impact on species' diel activities (Table 1).

Environmental variables included the crown closure (i.e., percent of ground covered by tree crowns) at each station and the fractional lunar phase during which the detection event was recorded (range: 0–1, where 0 = new moon and 1 = full moon), with the latter being calculated using R package *suncalc* (v 0.5.1) [58]. We did not evaluate crown closure within a buffer (as done for trail density and road density), because this variable was meant to characterize light availability in the immediate vicinity of the CT, therefore controlling for a habitat characteristic which might impact the likelihood of wildlife using a specific area at night. Previous studies have found the presence of other sympatric species may also impact diel activity patterns either independently, or in addition to human activity [37, 62]. However, we did not include the presence of other species as variables in our models for a number of reasons: 1) environmental conditions or anthropogenic influences which drive species distributions may lead to co-occurrences which could be incorrectly interpreted as interactions [63]; 2) there are additional species which we did not monitor (e.g., rodents, non-mammals) which might interact independently with each of our focal species and such interactions might produce artificially positive associations (e.g., coyotes and bobcats may be more nocturnal in areas with large numbers of nocturnal rodents independently of one another) [64, 65]; and 3) statistically accurate measures of co-occurrence require large sample sizes—a characteristic which our data set lacked, as evident in the distribution of detections of each species which occurred within the same time period of other species' detections (S2–S7 Figs) [66]. We acknowledge that the omission of species co-occurrence data may overlook interactions between predators or competitors occurring in our system, but given the numerous issues with modeling co-occurrences of species with our dataset, and our original goal of testing hypotheses about impacts of anthropogenic disturbance on wildlife diel activities, this mode of analysis was out of the scope of our research. Future work could build on our models with alternative analytical approaches such as avoidance-attraction ratios [43], causal modeling approaches (e.g., structural equation models) [67], or experimental manipulations [10] to determine whether the occurrence of other species may also play a role in shaping diel activity patterns of wildlife in this system.

Prior to model construction, all predictor variables were scaled by subtracting the mean and dividing by one standard deviation (SD) to compare relative effects of each variable on wildlife nocturnality, and all variables were tested for excessive multicollinearity (all Pearson's $r < 0.7$; S1 Fig). We constructed all models with flat priors (e.g., uniform distribution with bounds from -infinity to infinity), running them with 100,000 iterations across 4 chains (burn-in period = 5,000, thinning rate = 1) using the R package *brms* (v 2.18.0) [68]. We confirmed model convergence with the Gelman-Rubin statistic (R-hat $< 1.1$) [69] and by visually assessing trace plots. We also tested for spatial autocorrelation using Moran's I tests for each species with R package *spdep* (v 1.2–7) [70] (S1 Table). Once models converged, parameter estimates were considered to have moderate evidence of an effect on wildlife nocturnality if their 90% credible intervals did not include zero, and strong evidence of an effect on wildlife nocturnality if their 95% CIs did not include zero. The former criterion was included as a 90% CI that does not include zero corresponds to at least a 95% chance that the parameter estimate is entirely positive or negative (i.e., a one-sided 95% confidence interval), and similar levels of confidence have been utilized elsewhere [71, 72]. We caution that our interpretation of strength of evidence of an effect should not be conflated with strength of an effect, which we assessed with effect sizes (i.e., magnitude of mean parameter estimate). Additionally, we refrained from calculating p-values and instead focus on a gradient of strength of evidence to reflect a shift away from traditional "significance testing" [73]. To this end, we also discuss effects of variables relevant to our hypotheses which were not deemed strong or moderate, rather than just those selected on strength of evidence [74].

# Results

## Camera trap detections

From 23,928 camera trap-days of sampling effort, we collected over one million images representing 1,912 independent detection events of our six focal wildlife species. These included 46 cougar events, 290 black bear events, 709 black-tailed deer events, 248 snowshoe hare events, 416 coyote events, and 203 bobcat events (Table 2). Cougars and black-tailed deer maintained diel activity curves with mostly bimodal distributions, peaking at or near dawn and dusk (cougar activity peaked around 6:00 am and 9:00 pm, while black-tailed deer activity peaked at approximately 6:00 am and 6:00 pm; Fig 2). Black bear activity was greatest from approximately 6:00 am to 9:00 pm, with activity steadily increasing from around 2:00 pm to 9:00 pm where it peaked and subsequently dropped (Fig 2). Snowshoe hares, coyotes, and bobcats all showed nocturnal activity patterns, with snowshoe hare activity being greatest from approximately 10:00 pm to 4:00 am, coyote activity peaking around 6:00 am and 9:00 pm (and remaining active during nighttime hours), and bobcat activity peaking around 2:00 am (Fig 2). Accordingly, black bears appeared to be the least nocturnal of these species (mean nocturnality = 4.70 decimal hours from solar noon; Table 2), while snowshoe hares were the most nocturnal (mean nocturnality = 9.23 decimal hours from solar noon; Table 2). Nonetheless, all species were detected at least once within two decimal hours of solar noon, and at least once within one decimal hour of solar midnight (i.e., 12 decimal hours from solar noon), indicating some level of variation within all species' diel activity patterns was present. Additionally, we identified 111,468 independent detections of humans, 87.8% of which were detected between 9:00 am and 6:00 pm (S8 Fig).

## Temporal scale of human influence

Bayes factors associated with preliminary models indicated that variation in each of the six focal species' diel activity patterns were differentially explained by the three temporal scales (daily, weekly, and monthly) of direct human presence (S2 Table). Parameter estimates and CIs suggested most differences were minimal (S9 Fig). Nevertheless, variations in cougar and coyote nocturnalities were best explained by monthly human detection rates (Bayes factors: cougar = 25.69, coyote = 2937.28), black bear nocturnality was best explained by weekly human detection rates (Bayes factor = 4.30), and bobcat nocturnality was best explained by the number of humans detected during the same calendar day as the wildlife detection event (Bayes factor = 1.45). Subsequent models for each species therefore included these variables as measures of direct human presence. For black-tailed deer and snowshoe hares, no preliminary

**Table 2. Summary of detections acquired via camera trap in Golden Ears Provincial Park and University of British Columbia Malcolm Knapp Research Forest, BC.** For wildlife, the number of independent detections (number of observations) based on a 30-minute independence threshold, as well as the average and range of nocturnality values (decimal hours from solar noon). For humans, the number of independent detections used a 1-minute independence threshold.

| Species | No. independent detection events | Mean nocturnality (decimal hours from solar noon) | Range nocturnality (min.–max.) |
|---|---|---|---|
| Cougar | 46 | 6.47 | 0.50–11.20 |
| Black bear | 290 | 4.70 | 0.05–11.75 |
| Black-tailed deer | 709 | 5.74 | 0.03–11.86 |
| Snowshoe hare | 248 | 9.23 | 1.29–11.99 |
| Coyote | 416 | 7.27 | 0.01–11.98 |
| Bobcat | 203 | 8.11 | 0.22–11.99 |
| Human | 111,486 | - | - |

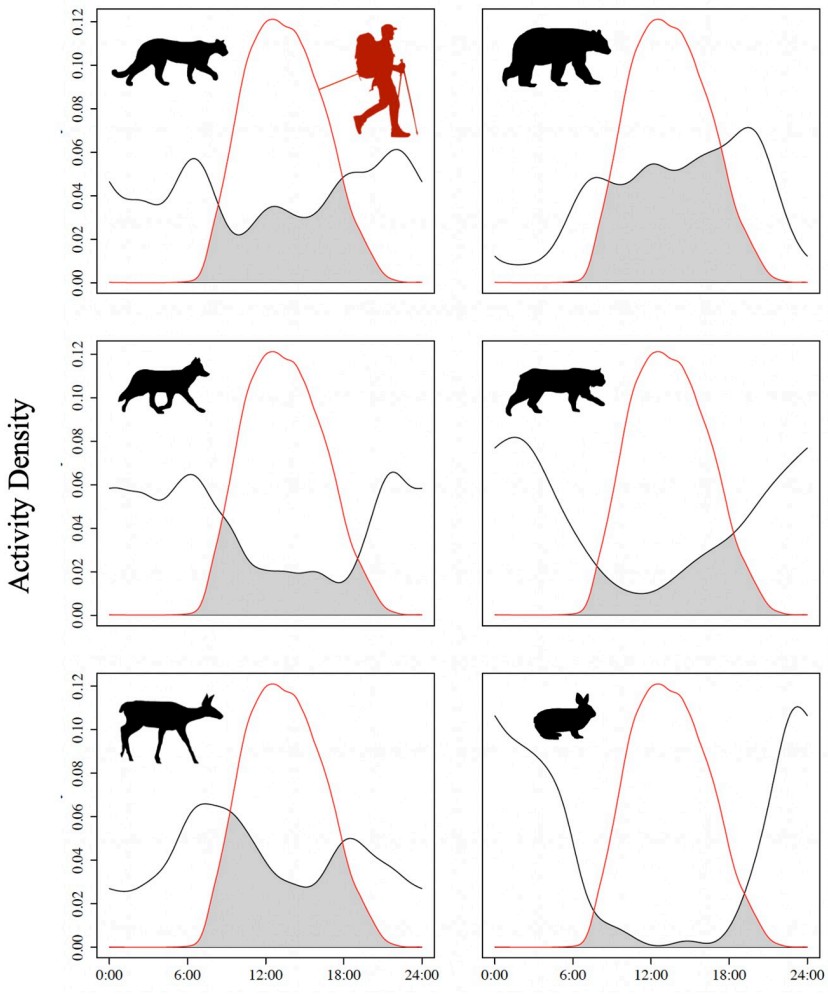

**Fig 2. Kernel density curves, which illustrate the times throughout the day each species (black curves) and humans (red curves) were detected at all camera stations in Golden Ears Provincial Park and the University of British Columbia Malcolm Knapp Research Forest, BC.** The x-axes show the time of day (hours), and the y-axes show the density of detections for each species. Species included cougars (top left), black bears (top right), coyotes (middle left), bobcats (middle right), black-tailed deer (bottom left), and snowshoe hares (bottom right). Kernel density estimates and curves were generated using R package *overlap* (v 0.3.4) [75].

model outperformed the intercept-only null model (all Bayes Factors < 1). Thus, subsequent models for these two species did not include a measure of direct human presence.

### Drivers of wildlife nocturnality

In our full models, we found no strong evidence that any species' nocturnality was impacted by direct human presence (all 95% CIs included zero; S3 Table). However, we found moderate evidence that black bears were more nocturnal in response to greater weekly human detection rates (mean posterior estimate = 0.35, 90% CI = 0.04 to 0.65; Fig 3). Furthermore, we identified strong evidence that coyotes were more nocturnal in regions of higher trail density (estimate = 0.81, 95% CI = 0.46 to 1.17), whereas snowshoe hares were less nocturnal in these areas (estimate = -0.87, 95% CI -1.29 to -0.46). We also found evidence that coyotes and cougars

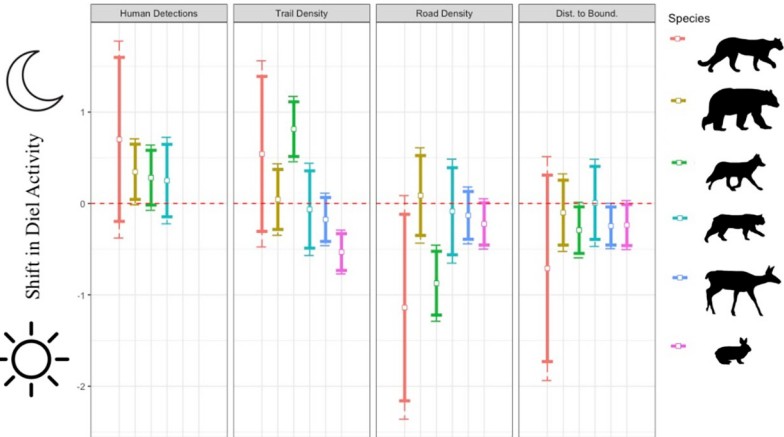

**Fig 3. Parameter estimates (center white point in each line), 90% credible intervals (thick error bars), and 95% credible intervals (thin error bars) from models which contrasted nocturnality (shift in decimal hours of a wildlife detection from solar noon) against measures of human disturbance.** Species (top to bottom in legend) include cougars, black bears, coyotes, bobcats, black-tailed deer, and snowshoe hares. Estimates above red dashed intercept (y = 0) indicate positive effects on wildlife nocturnality (e.g., shifts away from solar noon), while estimates below the intercept indicate negative effects (e.g., shifts towards solar noon). "Human detections" = number of daily human detections (bobcats), weekly human detection rate (black bears), or monthly human detection rate (cougars and coyotes), "Trail Density" = density of recreation trails within a 500 m buffer of the camera station (m/m$^2$), "Road Density" = density of roads within a 500 m buffer of the camera station (m/m$^2$), "Dist. to Bound." = distance from the camera trap to the urban-wildland boundary (m).

were less nocturnal in areas of higher road density (coyote estimate = -0.87, 95% CI = -1.29 to -0.46; cougar estimate = -1.14, 90% CI = -2.16 to -0.12). Additionally, coyotes, black-tailed deer, and snowshoe hares were all moderately more nocturnal in areas closer to the urban-wildland boundary (estimates and 90% CIs: coyote = -0.29, -0.55 to -0.04, black-tailed deer = -0.25, -0.45 to -0.04, snowshoe hare = -0.24, -0.46 to -0.01). For environmental effects, there was strong evidence suggesting that black bears were more nocturnal in areas with less crown closure (estimate = -0.35, 95% CI = -0.68 to -0.02), and coyotes were moderately more nocturnal in areas of higher crown closure (estimate = 0.27, 90% CI = 0.01 to 0.53) and during times of greater lunar illumination (estimate = 0.27, 90% CI = 0.03 to 0.51). Moran's I tests indicated a lack of spatial autocorrelation in all models with the exception of the coyote model (Moran's I = 0.25, p = 0.04).

## Discussion

### Varied effects of human disturbance on nocturnality

We did not identify strong evidence for effects of direct human presence on wildlife nocturnality, but found moderate evidence that greater direct human presence may increase black bear nocturnality. Despite generally weak effects, all predator and mesopredator species showed the predicted direction of predominantly greater nocturnality in response to direct human presence. Furthermore, we found strong effects of infrastructure on coyote and snowshoe hare nocturnality, with coyotes being more nocturnal in regions of higher trail density and snowshoe hares maintaining an opposite relationship with trail density. However, we also identified spatial autocorrelation in the coyote model, implying there may have been non-independence in coyote behavior across nearby sampling locations. We also found moderate evidence of cougars, coyotes, black-tailed deer, and snowshoe hares altering their diel activity patterns in response to infrastructure, plus both strong and moderate evidence for black bear and coyote

diel activities shifting with environmental factors, reinforcing the importance of controlling for alternative sources of variation when investigating these responses.

Our finding that black bears were moderately more nocturnal in response to greater direct human presence suggests black bear behavior may be most influenced by human use of the landscape. Prior studies have indicated that black bears may be displaced by recreation [43], and/or vehicles used in research or forest management operations [44], but these studies characterized displacement differently (e.g., attraction-avoidance ratios, weekly habitat use). Little research has reported impacts of human activity on black bear diel activities, and of those which did, black bears were found to be more nocturnal at urban interface areas [50], to adjust their diel activities in response to seasonality [76], and/or to maintain mostly crepuscular activity patterns regardless of external influences [77]. Therefore, we provide a novel account of black bears being more nocturnal in response to direct measures of human activity. We speculate this behavior may be due to the increased risk of wildlife destruction that black bears face in areas of higher human influence [50], which thereby imposes stronger incentive to reduce temporal overlap with humans. Likewise, since black bears were the least nocturnal of the species we considered (Fig 2), they may be more likely to shift their temporal niche to accommodate primarily diurnal humans relative to other species that maintain more natural temporal niche segregation. To this end, other species in this system may have circadian rhythms that naturally allow for temporal segregation from humans, and therefore may not need to shift in response to human pressures. However, additional research would be needed to confirm whether these species are indeed nocturnal in areas without human presence. Some species which are typically thought of as nocturnal or crepuscular may exhibit diurnal behavior in areas where humans or predators are excluded [51, 78], so there is also a need for additional monitoring in areas without humans to understand the full range of wildlife diel activities both with and without human influences.

We also identified strong evidence that coyotes were more nocturnal in areas of greater recreation trail density, whereas snowshoe hares were less nocturnal in these spaces. An increase in coyote nocturnality in regions of higher trail density may provide temporal refuge for snowshoe hares to use these spaces more readily during daytime hours. Research has outlined the capacity for prey species to exploit such refugia [79], but snowshoe hares have previously been identified as primarily nocturnal regardless of external factors [39] and we found no prior studies reporting shifts in snowshoe hare diel activities in response to human disturbance. Hence, we provide a unique instance of snowshoe hares responding to infrastructure which facilitates human activity, potentially in and of themselves, or as a repercussion of contrasting behavioral adaptations of a sympatric carnivore. Coyotes were also less nocturnal in areas of greater road density, and may therefore adjust their behavior to efficiently navigate a landscape of myriad disturbances using less heavily recreated areas of higher road density (e.g., Malcolm Knapp) during the day and more heavily recreated areas of higher trail density (e.g., Golden Ears) at night. This potential shift in use between areas could potentially explain the spatial autocorrelation detected in the coyote model; we recommend additional modeling to more effectively understand the drivers of coyotes diel activities in this system. Prior research has indicated coyotes may be more nocturnal in areas of higher human activity, but that this relationship may not hold true when considering landscape features which imply greater human presence (e.g., areas of greater human footprint) [28]. Thus, our findings may contrast previous work, as we identified strong evidence of coyote diel activities shifting with road or trail density, but no clear patterns of coyote nocturnality responding to direct human presence. Overall, these discrepancies emphasize that the study of wildlife diel activity patterns is still developing [80], and that future work should consider impacts of both direct human presence and landscape features associated with greater implied human activity when possible.

We also identified a few infrastructure variables which had moderate impacts on the diel activities of a number of species. Namely, coyotes, black-tailed deer, and snowshoe hares were all more nocturnal in areas closer to the urban-wildland boundary. These results support prior work which asserts a variety of wildlife species are more nocturnal in areas of higher human influence [27], and suggests that although these species are known to efficiently utilize urban spaces [81], they may do so more often during nighttime hours. Future work should monitor residential and agricultural areas adjacent to the PA in order to determine whether species selectively use these areas during dawn, dusk, or nighttime hours, retreating back to the PA during the day. Such a finding could bolster support for PA establishment, maintaining that some species can evade disturbance in PA-adjacent lands by altering their temporal activity patterns to utilize PA-adjacent lands at night, and nearby PAs during daytime hours. We also found moderate evidence that cougars were less nocturnal in areas of higher road density. Cougars are known to avoid paved roads, while potentially selecting for dirt roads of lower use [82] similar to those found in Malcolm Knapp. Prior studies regarding cougar temporal activity patterns have also noted cougars are often more nocturnal in response to humans [28, 49], and others have posited cougars may use logging roads more often in the evening in areas where humans are present [83]. Roads in Malcolm Knapp are not as heavily used as roads or recreation trails in Golden Ears (Golden Ears averaged 3728 humans per camera station over the entire study period while Malcolm Knapp only averaged 272). Therefore, cougars may traverse roads in Malcolm Knapp to facilitate easier movement, and may do so during daytime hours due to the lack of perceived threats from human activity.

## Broader patterns and implications of wildlife nocturnality

While we identified no strong or moderate evidence for effects of direct human presence on wildlife nocturnality except for black bears, all estimates of predator and mesopredator species' nocturnality in response to direct human presence variables were predominantly positive (Fig 3). The road to Golden Ears is gated at night, barring people without a camping permit from easily accessing most recreation trails at dawn or dusk. Therefore, recreation in the park is predictably diurnal, and pressure on species to become completely nocturnal might not be as great as in landscapes where hikers can more readily access trails during crepuscular hours. Accordingly, reduced human activity during crepuscular hours may allow species to maintain adequate segregation from humans by being more active during these times, rather than becoming entirely nocturnal. Consequently, such smaller-scale shifts in diel activities might translate to smaller effect sizes, and might therefore be less likely to emerge as "strong" predictors [84]. Similar lesser shifts towards crepuscular activity, rather than complete shifts towards nocturnality, have been identified elsewhere [85], and may provide support for restrictions on recreational use of trails to certain hours of the day in areas where shifts in wildlife diel activities have negative repercussions [31]. Accordingly, it may be adequate for PA managers to consider "temporal refugia", rather than complete restrictions on human activity (i.e., trail closures) when attempting to reduce impacts of recreation on wildlife.

Additionally, the lack of strong or moderate evidence for effects of direct human presence on wildlife nocturnality could be due to behavioral variation within species, which may imply a level of habituation in some individuals. This could have pressing implications, especially for human-carnivore coexistence [49]. Some individuals in the landscape may be habituated to human activity and landscape features which facilitate human activity, while others still maintain a natural fear response to humans, therefore leading to greater uncertainty in effects estimated at the population level (i.e., across these diverse individuals). Individual behavioral variation is common in predators, and thus, individual-targeting approaches are sometimes

recommended for conflict management [86]. Future work could employ methods which can more effectively reveal individual variation (e.g., GPS collars) to understand if certain individuals are showing signs of habituation, lacking a fear response to humans that would otherwise drive increased nocturnality and hence coexistence through temporal segregation, instead increasing the risk of temporal overlap and conflict.

While recent studies have emphasized widespread nocturnality is increasingly prevalent in regions of higher human activity, those studies are often large in spatial or temporal resolution and therefore require methods which can not reveal small-scale shifts in diel activity patterns. For instance, Gaynor et al. [27] noted prevalent global trends in wildlife species becoming more nocturnal in regions of higher human influence. However, due to their use of multiple sources of data, and the difficulty in standardizing across such disparate sources, they used a coarse nocturnality measure of day/night rather than a continuous measure of temporal activity. This sort of analysis may interpret smaller-scale shifts in wildlife diel activity (e.g., species becoming more crepuscular) as full shifts towards nighttime activity. Gaynor et al. [27] also utilized a categorical designation of human use (high vs. low use)—again due to the large scale of the study—but this coarse-scale characterization of human activity does not reveal how different magnitudes of human activity might impact wildlife behavior. At a global scale, such coarse methods are necessary due to a scarcity of finer-scale data across large spatial extents. However, we speculate that future efforts to understand wildlife nocturnality as it relates to human activity at a global scale might leverage global networks of wildlife monitoring instruments (e.g., camera traps, autonomous recording units) to obtain continuous data on both human and wildlife activities, along with policies which promote open access science, to better develop a more widespread understanding of trends in wildlife behavior [87].

Finally, the extent to which PAs can promote human-wildlife coexistence in the face of growing levels of visitation remains a topic of great concern. We present evidence that black bears may temporally segregate directly from primarily diurnal recreationists in PAs, but that landscape features which facilitate recreation likely impact wildlife diel activities to a greater degree. These findings regarding wildlife responses to infrastructure are consistent with previous works [26, 28], and prompt questions regarding the effectiveness of PAs in mitigating human impacts on wildlife. Ultimately, our work suggests PA management should strongly consider limiting (or carefully planning) new infrastructure in parks to ensure minimal disturbance on species inhabiting these landscapes. Furthermore, fine-scale temporal segregation such as altered diel activity patterns in light of human influences may be regarded as a mechanism by which humans and wildlife can coexist [22]. However, it remains unclear whether such shifts might have negative impacts (i.e., risk effects to species which avoid certain niches [2]). In some cases, human activity may fundamentally alter trophic structures and natural predator-prey dynamics by restricting the temporal niches predators and prey may occupy, thereby preventing prey species from avoiding predators [36]. Other risk effects may include lower reproductive output [34, 35] or reduced feeding time [11]. Nevertheless, long-term camera trapping studies have also shown increasing occupancy of high human use areas by of a number of wildlife species when they are able to adequately segregate via increased nocturnality [88]. Therefore, it is unclear whether shifts in wildlife diel activities in response to human activity are predominantly negative or positive, reinforcing a necessity for future work to continue investigating how wildlife nocturnality changes with increasing human pressures.

## Conclusions

Wildlife nocturnality is commonly thought to increase in the face of greater human activity. However, the characterization of these phenomena hinges on the selection of appropriate

variables, and methodological approaches which can reveal varied levels of detail about these relationships. We used CTs to understand how wildlife and humans share time in a PA and adjacent university-managed research forest, presenting evidence that some species show strong diel activity responses to infrastructure, but little evidence that direct human presence impacts these activity patterns. Nevertheless, we found that black bears may show moderate temporal avoidance of humans using the area, and posit that the lack of effects for other species may be due to smaller-scale shifts towards crepuscular activity rather than fully nocturnal behavior, or wide variation in individual behaviors. We encourage future studies to include as much detail as possible (e.g., use continuous measures of both human and wildlife activity if available) when investigating how wildlife temporally segregate from human activity, and we anticipate increases in open science (e.g., data availability) will facilitate the use of greater levels of detail in large-scale efforts to understand these patterns. We emphasize that understanding these relationships is crucial, as wildlife temporal segregation may serve as a mechanism promoting human-wildlife coexistence, but it may also facilitate negative shifts in community trophic structure, or risk effects which reduce the overall fitness of wildlife populations. While wildlife in PAs may be subject to human disturbance through recreation or infrastructure, the extent to which these factors impact wildlife in space or time is yet unclear, accentuating a need for further investigation which could further explore how the two interact to promote biodiversity conservation broadly.

## Supporting information

**S1 Fig. Multicollinearity (tested by Pearson's correlation coefficient) of predictor variables used in Bayesian GLMs which regressed wildlife nocturnality against human detections, landscape variables which represented potential human influences (e.g., trail density), and environmental variables which might impact wildlife diel activities (e.g., lunar phase).** Species included cougars (top left), black bears (top right), coyotes (middle left), bobcats (middle right), black-tailed deer (bottom left), and snowshoe hares (bottom right). For all species, daily human detections, and weekly and monthly human detection rates were highly collinear (all R > |0.7|; min. = 0.76). Maximum correlation values aside from human detection variables varied by species, but all were < |0.7| (max. cougar R = -0.51, crown closure and trail density; max. black bear R = -0.58, crown closure and trail density; max. black-tailed deer R = 0.61, crown closure and lunar fraction, max. snowshoe hare R = -0.45, crown closure and trail density; max. coyote R = 0.53, crown closure and lunar fraction; and max. bobcat R = 0.48, crown closure and lunar fraction).
(PNG)

**S2 Fig. Histograms showing the distribution of detection data for all non-cougar species at three temporal scales for the cougar detection dataset.** The cougar dataset comprises all independent detection events of cougars. Therefore, histograms show the detections of each species during that same day of the detection event (left column), the weekly detection rate during that same week of the detection event (middle column), and the monthly detection rate during that same month of the detection event (right column). Non-cougar species are black bears, black-tailed deer, snowshoe hares, coyotes, and bobcats (rows, from top to bottom).
(PNG)

**S3 Fig. Histograms showing the distribution of detection data for all non-black bear species at three temporal scales for the black bear detection dataset.** The black bear dataset comprises all independent detection events of black bears. Therefore, histograms show the detections of each species during that same day of the detection event (left column), the weekly

detection rate during that same week of the detection event (middle column), and the monthly detection rate during that same month of the detection event (right column). Non-black bear species are cougars, black-tailed deer, snowshoe hares, coyotes, and bobcats (rows, from top to bottom).
(PNG)

**S4 Fig. Histograms showing the distribution of detection data for all non-black-tailed deer species at three temporal scales for the black-tailed deer detection dataset.** The black-tailed deer dataset comprises all independent detection events of black-tailed deer. Therefore, histograms show the detections of each species during that same day of the detection event (left column), the weekly detection rate during that same week of the detection event (middle column), and the monthly detection rate during that same month of the detection event (right column). Non-black-tailed deer species are cougars, black bears, snowshoe hares, coyotes, and bobcats (rows, from top to bottom).
(PNG)

**S5 Fig. Histograms showing the distribution of detection data for all non-snowshoe hare species at three temporal scales for the snowshoe hare detection dataset.** The snowshoe hare dataset comprises all independent detection events of snowshoe hares. Therefore, histograms show the detections of each species during that same day of the detection event (left column), the weekly detection rate during that same week of the detection event (middle column), and the monthly detection rate during that same month of the detection event (right column). Non-snowshoe hare species are cougars, black bears, black-tailed deer, coyotes, and bobcats (rows, from top to bottom).
(PNG)

**S6 Fig. Histograms showing the distribution of detection data for all non-coyote species at three temporal scales for the coyote detection dataset.** The coyote dataset comprises all independent detection events of coyotes. Therefore, histograms show the detections of each species during that same day of the detection event (left column), the weekly detection rate during that same week of the detection event (middle column), and the monthly detection rate during that same month of the detection event (right column). Non-coyote species are cougars, black bears, black-tailed deer, snowshoe hares, and bobcats (rows, from top to bottom).
(PNG)

**S7 Fig. Histograms showing the distribution of detection data for all non-bobcat species at three temporal scales for the bobcat detection dataset.** The bobcat dataset comprises all independent detection events of bobcats. Therefore, histograms show the detections of each species during that same day of the detection event (left column), the weekly detection rate during that same week of the detection event (middle column), and the monthly detection rate during that same month of the detection event (right column). Non-bobcat species are cougars, black bears, black-tailed deer, snowshoe hares, and coyotes (rows, from top to bottom).
(PNG)

**S8 Fig. Kernel density plot showing the diel activities of humans detected per month in Golden Ears Provincial Park and University of British Columbia Malcolm Knapp Research Forest, BC.** Each line represents the average kernel density (y-axis) of human detections throughout a 24-hour period (x-axis) for each month of camera trap sampling.
(PNG)

**S9 Fig. Model results from preliminary univariate models, which tested whether daily human detections, weekly human detection rates, or monthly human detection rates best**

**explained variation in species' nocturnalities.** Each panel corresponds to a specific species, x-axes show different human-use variables included in univariate models, while y-axes show disturbance effect (parameter estimates). Center points in each line represent mean parameter estimates, while lines represent 95% credible intervals.
(PNG)

**S1 Table. Results of testing for spatial autocorrelation with Moran's I tests.** All species' nocturnality data showed no signal of spatial autocorrelation, with the exception of coyotes ($p < 0.05$).
(DOCX)

**S2 Table. Bayes factors associated with the three preliminary models constructed for each species.** These models contrasted wildlife nocturnality against 1) the number of humans detected during the day of the wildlife detection event (i.e., "daily human detections"), 2) the number of humans detected throughout the week of the wildlife detection event divided by the number of days the camera was active (i.e., "weekly human detection rate"), and 3) the number of humans detected throughout the month of the wildlife detection event (i.e., "monthly human detection rate"). For each species, the model with the greatest Bayes Factor (bolded) was assumed to best explain variation in the data and was therefore used in construction of subsequent models. If no model outperformed the intercept-only model for a given species (all model Bayes Factors < 1), the species was modeled without a measure of direct human activity.
(DOCX)

**S3 Table. Parameter estimates, 95% credible intervals, and 90% credible intervals from Bayesian regression models which contrasted wildlife nocturnality against measures of human activity, and landscape measures of implied human activity, while controlling for alternative factors (e.g., environmental variables) which might also influence wildlife diel activities.** [a] "Dist. to bound" = Distance to the urban-wildland boundary (m).
(DOCX)

## Acknowledgments

We thank members of the Wildlife Coexistence Lab at The University of British Columbia for assistance, in particular: Mitch Fennell, Taylor Justason, Katie Tjaden-McClement, Isla Francis, and Gerlissa Chan for help in deploying and checking camera sites; Katie Tjaden-McClement, Isla Francis, and Gerlissa Chan for help classifying camera trap photos; and Catherine Sun, Chris Beirne, and Alys Granados for comments on preliminary drafts and analytical considerations. We thank Melanie Percy, James Quayle, Sam Stickney, Joanna Hirner, Daris Lapointe, Simon Debisschop, and Gareth Wheatley from BC Parks for assistance and support. Permission to conduct research in Malcolm Knapp Research Forest was granted by research coordinator Ionut Aron.

## Author Contributions

**Conceptualization:** Michael Procko, A. Cole Burton.

**Data curation:** Michael Procko.

**Formal analysis:** Michael Procko.

**Funding acquisition:** A. Cole Burton.

**Investigation:** Michael Procko.

**Methodology:** Michael Procko, Robin Naidoo, Valerie LeMay, A. Cole Burton.

**Project administration:** Michael Procko, A. Cole Burton.

**Supervision:** Michael Procko, Robin Naidoo, Valerie LeMay, A. Cole Burton.

**Validation:** Michael Procko.

**Visualization:** Michael Procko.

**Writing – original draft:** Michael Procko.

**Writing – review & editing:** Michael Procko, Robin Naidoo, Valerie LeMay, A. Cole Burton.

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
