## [Decision Letter · Decision Letter 0]

22 Dec 2022

PONE-D-22-31729Human presence and infrastructure impact wildlife nocturnality differentially across an assemblage of mammalian speciesPLOS ONE

Dear Dr. Procko,

Thank you for submitting your manuscript to PLOS ONE. After careful consideration, we feel that it has merit but does not fully meet PLOS ONE’s publication criteria as it currently stands. Therefore, we invite you to submit a revised version of the manuscript that addresses the points raised during the review process.

We look forward to receiving your revised manuscript.

Kind regards,

Francesco Rovero, Ph.D.

Academic Editor

PLOS ONE

“This research was supported by the British Columbia Ministry of Environment and Climate Change Strategy (BC Parks) under agreements TP20JHQ024 and TP21JHQ013. Additional funding was received through grants to ACB from the Natural Sciences and Engineering Council of Canada and the Canada Research Chairs program, as well as from the University of British Columbia’s Faculty of Forestry.”

Additional Editor Comments:

I have received reviews from two referees. They both found your manuscript interesting and solid, and while they gave a very positive evaluation overall, they also provide a number of comments for improving it further and clarify some parts, especially on methods. 

Reviewers' comments:

Reviewer's Responses to Questions

**Comments to the Author**

1. Is the manuscript technically sound, and do the data support the conclusions?

Reviewer #1: Yes

Reviewer #2: Yes

2. Has the statistical analysis been performed appropriately and rigorously? 

Reviewer #1: Yes

Reviewer #2: Yes

3. Have the authors made all data underlying the findings in their manuscript fully available?

Reviewer #1: Yes

Reviewer #2: Yes

4. Is the manuscript presented in an intelligible fashion and written in standard English?

Reviewer #1: Yes

Reviewer #2: Yes

5. Review Comments to the Author

Reviewer #1: This is an interesting study that used camera trapping data in an original way to evaluate the possible influence of human presence on nocturnality of six mammal species in a temperate ecosystem. The study assessed the effect on nocturnality of both direct human presence and human infrastructures. Results are relevant and interesting, and they are fairly discussed with a proper approach to the Bayesian framework used for the analyses. Furthermore, the results have possible, and correctly stated, conservation implications targeting the management of human activities in protected areas.

Overall, the manuscript is well written, and I think it meets the rigorous standards for publication in the journal. However, I must point out that despite the data support the conclusions drawn by the authors, there is a point that remains unclear to me, and other revisions are required.

The main critic I move to the manuscript concerns the choice of avoiding biotic variables in modelling nocturnality. The manuscript focuses on human presence and its influence on nocturnality with specific hypotheses, but it remains unclear to me why abiotic factors such as lunar phase and forest cover were (correctly) considered as potentially influencing the diel activity patterns of mammals (as confounding factors, I suppose; they are called “alternative factors” in L724), whereas biotic factors (e.g., the presence of competitors or predators) were not considered (see Bonnot et al. 2019, Journal of Animal Ecology for an example). For instance, the study area hosts four carnivores of different sizes and their spatio-temporal interaction (even pairwise) could have been influential (e.g., what happens to other predators’ nocturnality in areas of higher bear detection?); the same could be true for predator-prey dynamics such as snowshoe hare-coyote/bobcat and black-tailed deer-black bear/cougar, just to give some examples. The need “to consider a number of factors when investigating anthropogenic effects on wildlife nocturnality” is clearly stated in the manuscript (L83-84, and also L356-357), and for this reason I would like to ask for further details about this choice.

Specific comments:

Abstract L32-33: I suggest caution in using the word “wildlife” when stating that the findings of the study imply a stronger influence of human features compared to direct human presence. For instance, the study focused on medium to large sized mammals (with the exception of the hare), thus the possibly differential impact on other small-sized mammals, such as small carnivores (e.g., mustelids) or micro-mammals, is not known, and such a generalization is potentially misleading.

L56: I think “human presence” is too general and its use should be limited to cases in which both human activities and human infrastructures are considered together. I suggest sticking to the definition of “direct human presence” used in Abstract L33 to properly maintain the two categories of human presence separated. The same is valid for all the other cases in the manuscript (e.g., line 106, 186, 206, 305, 314 etc.).

L70: human presence not only affects predator-prey interactions, but it could also affect temporal segregation among predators, thus possibly increasing intra-guild competition, see Wang et al. 2015, Biological Conservation for an example.

L93: I understand the emphasis on using continuous measurements of time, but I would suggest using “coarse” instead of “coarser”, as time of the day could be categorized in day, night and twilight, and in this case, with enough data, it could still be possible to properly detect crepuscular peaks in activity.

L110: I do not like the use of “prevalent” in stating the first main hypothesis, as this term is never found again in the manuscript when dealing with the results and their implications for that specific hypothesis. I would suggest to avoid that specific word to remain more coherent with the rest of the manuscript.

L113-115: I suggest sticking to the singular form when presenting the species names for the first time, especially when associated with their scientific name, as the reference is to species in the widest sense of their definition.

L130-131: This part of the manuscript links with my main critic, as the study area is not predator-free. For this reason, black-tailed deer might have shifted its diel activity pattern towards crepuscular periods to avoid humans, or predators, or even both. I pointed this out also to help answering my main critic.

L142: I suggest to include an explication of the BC abbreviation for British Columbia, and to also include the country (Canada). I think it is important to give all the information a reader could need to properly understand the study (not all people around the world may know what or where BC is).

L184: Information about data extraction from camera trap images is missing.: software used for image managing (if any), and specific R packages or software used for event extraction (if any).

L200: The R package name should be accompanied by the package version. This is valid for all packages named through the manuscript.

L208-212: I suggest sticking to the definition of weekly/monthly detection RATES correctly used in other parts of the manuscript (e.g., Table 1), as they are number of detections divided by the camera trapping effort. Furthermore, the definition given does not correspond to the one given in the TABLE S1 FILE caption, where it is stated that the variable refers to the detection rate for the week of the specific camera trap event. Please clarify this point and make it coherent throughout the manuscript. See also the associated comment for lines 700-702.

L235: It is unclear to me why human infrastructure variables were evaluated in a buffer, whereas crown closure was considered only for the camera trapping site. I would like to ask for further details about this choice.

L240-243: Information about eventual adaptation/warm-up phase and thinning are missing. Furthermore, I suggest to explicitly state the threshold used in evaluating convergence through the Gelman-Rubin Rhat.

L255: as for lines 208-212, consider that the ones within the day are detections, the ones in the week or month are detection rates.

L259 TAB1: “to THE anticipated”

L290 TAB: if I understood well, this should be table TWO, not table one.

L299 FIG2: “The x-axes […] and y-axes”, whether using the article or not, I suggest doing it coherently in this case.

L307; L309: again, those are detection rates.

L321; L377: I suggest using whereas instead of while in these case.

L401: “utilize urban areas”, remove the A article.

L453: “A continuous measure”

L478: “the temporal niche IN which both predators and prey occupy”, this statement is unclear to me; please provide further explanation or rephrase it.

L700-702 TAB S1: It is unclear to me why you refer to “the average” detection rate for weeks and months in this case, whereas no reference to the average value of detection rates was made throughout the manuscript.

Reviewer #2: Review of “Human disturbance and infrastructure impact wildlife nocturnality differentially across an assemblage of mammalian species”

The manuscript describes a research project focused on assessing the effect of human active potential disturbance and infrastructure presence on a set of mammalian species along a gradient of human footprint in western Canada.

I found the manuscript interesting and well written. I list my comments and suggestions below:

Introduction and Discussion are well written and interesting.

Line 46: Instead of saying that ‘PAs are often thought to provide refuge..’ I would rather say that PAs should provide refuge from anthropogenic pressures, since we established PAs for that very reason, and if they do not offer any refuge from anthropogenic pressures then they are failing to fulfil their scope. I suggest you should deepen this point a little bit in this part of the Intro.

Lines 64-67: Perhaps I misunderstood your point here, but what do you mean exactly when you say that different species may respond to human disturbance at different temporal scales? Why would fine-scale changes in human presence not affect species that avoid seasonal or monthly patterns in human disturbance? Is there any evidence of animals responding to e.g. increase in tourism in summer not responding to daily changes in human activity?

Line 73: you already used the verb underscore at line 56.

Lines 76-77: Is this ability really linked positively to body-mass? I would say is rather typical of generalist species, that can exploit disturbed areas by shifting to nocturnality, while more sensitive species usually avoid disturbed areas also spatially, or spatio-temporally and not only temporally.

Line 92: Isn’t wildlife an uncountable noun? I think ‘wildlife is active’ is the correct form.

Line 95: Concerning human disturbance with social media data I think this research could be very useful here: Corradini, A., Randles, M., Pedrotti, L., van Loon, E., Passoni, G., Oberosler, V., ... & Cagnacci, F. (2021). Effects of cumulated outdoor activity on wildlife habitat use. Biological Conservation, 253, 108818. https://doi.org/10.1016/j.biocon.2020.108818

Line 112: I find the term ‘human-related infrastructure’ odd, isn’t infrastructure intrinsically human related?

Lines 141-147: When describing the study area you do not mention that British Columbia is part of Canada, nor in which part of Canada it is located. Readers outside north America might not know where British Columbia is, nor what the acronym BC stands for.

Lines 153: I think you wanted to begin this sentence with ‘as a’ and not with ‘a’.

Line 163: I would erase the word ‘shaping’.

Lines 170-176: Here comes my main concern regarding your study: the sampling design is not based on a regular grid, thus cameras are set at very different distances one another. Sampling sites within Malcolm Knapp research forest are much closer compared to those within Golden Ears, and this may bias your results. Most probably sites in Malcolm Knapp will have a much higher level of spatial autocorrelation than those in Golden Ears. One way to overcome this would be to test if results of neighbouring sites are more correlated than those of sites that are more far apart (as for example done here: Kolowski, J. M., Oley, J., & McShea, W. J. (2021). High‐density camera trap grid reveals lack of consistency in detection and capture rates across space and time. Ecosphere, 12(2), e03350. https://doi.org/10.1002/ecs2.3350).

Line 195: You now mention ‘shifts’ for the first time, and it is not very clear what you refer to here. More in general, I am not sure that shift is the right word to use for your variable, since it seems to indicate a deviation or change from a certain, known value, whereas you are indicating a temporal distance from noon, taken as a reference.

Lines 205-212: I think that here it is not well justified and explained why there would be a need to test human passage at these three temporal scales. I imagine that sites with high overall human passage also have high weekly and daily values. The only situation in which this does not hold would be if human frequentation was concentrated in a few days and scarce for the rest of the sampling period (is that the case for your study areas?), but this possibility can easily been tested beforehand. Also, if that was the case, could you please list how would mammalian species respond differently to disturbance occurring at these three temporal intervals, are there evidence and example from the literature?

Line 228: I think there should be a short explanation of what Bayes Factor factor is for readers that are not familiar with it.

Results section: Have you assessed if you get similar results using a different nocturnality index (as for example nocturnal/(diurnal+nocturnal) events ratio)? I wonder whether the choice of the response variable is affecting your analysis.

Line 285-287: This sentence is not well linked to what is written before and after. Why is this observation important, what does it entail?

Line 364: This likewise here confounds me a little, you jump from effects of urban areas to seasonality too abruptly.

Line 401: erase the ‘a’ from ‘a urban spaces’.

Lines 413-415: Did you explicitly test whether roads and trails were more used in Golden Ears than Malcolm Knapp? It would be useful to report numerical output of such test here.

6. PLOS authors have the option to publish the peer review history of their article (what does this mean?). If published, this will include your full peer review and any attached files.

Reviewer #1: No

Reviewer #2: **Yes: **Marco Salvatori

---

## [Author Response · Author response to Decision Letter 0]

1 Mar 2023

Editor, journal requirements:

o We have ensured that our formatting meets the style requirements, including those for file naming. 

• Please state what role the funders took in study. If the funders had no role, please state: “The funders had no role in study design, data collection and analysis, decision to publish, or preparation of the manuscript.”

o We have included a revised financial disclosure in our cover letter.

• We note that you have stated that you will provide repository information for your data at acceptance. Should your manuscript be accepted for publication, we will hold it until you provide the relevant accession numbers or DOIs necessary to access your data.

o The following DOI holds the data in a private repository linked to the manuscript number (PONE-D-22-31729). Upon manuscript publication, access to the DOI will be changed to public automatically.

https://doi.org/10.5061/dryad.w3r2280v7

• Please include your full ethics statement in the ‘Methods’ section of your manuscript file. In your statement, please include the full name of the IRB or ethics committee who approved or waived your study, as well as whether or not you obtained informed written or verbal consent. If consent was waived for your study, please include this information in your statement as well.

o We have added our ethics statement to our methods section at lines 179-182:

“Written consent was obtained for all research undertaken, provided under protocol A18-0234 from the University of British Columbia’s Animal Care Committee and under protocol H21-01424 from the University of British Columbia’s Behavioural Research Ethics Board.”

• We note that Figure 1 in your submission contain [map/satellite] images which may be copyrighted. All PLOS content is published under the Creative Commons Attribution License (CC BY 4.0), which means that the manuscript, images, and Supporting Information files will be freely available online, and any third party is permitted to access, download, copy, distribute, and use these materials in any way, even commercially, with proper attribution. For these reasons, we cannot publish previously copyrighted maps or satellite images created using proprietary data, such as Google software (GoogleMaps, Street View, and Earth). 

If you are unable to obtain permission from the original copyright holder to publish these figures under the CC BY 4.0 license or if the copyright holder’s requirements are incompatible with the CC BY 4.0 license, please either i) remove the figure or ii) supply a replacement figure that complies with the CC BY 4.0 license. Please check copyright information on all replacement figures and update the figure caption with source information. If applicable, please specify in the figure caption text when a figure is similar but not identical to the original image and is therefore for illustrative purposes only.

o We have created a replacement Figure 1, including only freely available data which comply with the CC BY 4.0 license.

We have also updated the figure 1 caption (lines 210-233) to reflect this new figure, including all sources for the data used.

Reviewer #1, general comments:

• The main critic I move to the manuscript concerns the choice of avoiding biotic variables in modelling nocturnality. The manuscript focuses on human presence and its influence on nocturnality with specific hypotheses, but it remains unclear to me why abiotic factors such as lunar phase and forest cover were (correctly) considered as potentially influencing the diel activity patterns of mammals (as confounding factors, I suppose; they are called “alternative factors” in L724), whereas biotic factors (e.g., the presence of competitors or predators) were not considered (see Bonnot et al. 2019, Journal of Animal Ecology for an example). For instance, the study area hosts four carnivores of different sizes and their spatio-temporal interaction (even pairwise) could have been influential (e.g., what happens to other predators’ nocturnality in areas of higher bear detection?); the same could be true for predator-prey dynamics such as snowshoe hare-coyote/bobcat and black-tailed deer-black bear/cougar, just to give some examples. The need “to consider a number of factors when investigating anthropogenic effects on wildlife nocturnality” is clearly stated in the manuscript (L83-84, and also L356-357), and for this reason I would like to ask for further details about this choice.

o We acknowledge that the lack of biotic variables (e.g., the presence of interacting species) in our models is the biggest point of concern for this reviewer. While this critique is certainly understandable, we contend that inclusion of such co-occurrence data may not provide insight into interactions among species for a number of reasons, and that another modeling framework would likely be required to investigate this. We have added the following rationale for excluding this aspect from our modeling framework at lines 345-365.

“Previous studies have found the presence of other sympatric species may also impact diel activity patterns either independently, or in addition to human activity (37,62). However, we did not include the presence of other species as variables in our models for a number of reasons: 1) environmental conditions or anthropogenic influences which drive species distributions may lead to co-occurrences which could be incorrectly interpreted as interactions (63); 2) there are additional species which we did not monitor (e.g., rodents, non-mammals) which might interact independently with each of our focal species and such interactions might produce artificially positive associations (e.g., coyotes and bobcats may be more nocturnal in areas with large numbers of nocturnal rodents independently of one another) (64,65); and 3) statistically accurate measures of co-occurrence require large sample sizes—a characteristic which our data set lacked, as evident in the distribution of detections of each species which occurred within the same time period of other species’ detections (S2-S7 Figs.) (66). We acknowledge that the omission of species co-occurrence data may overlook interactions between predators or competitors occurring in our system, but given the numerous issues with modeling co-occurrences of species with our dataset, and our original goal of testing hypotheses about impacts of anthropogenic disturbance on wildlife diel activities, this mode of analysis was out of the scope of our research. Future work could build on our models with alternative analytical approaches such as avoidance-attraction ratios (43), causal modeling approaches (e.g., structural equation models) (67), or experimental manipulations (10) to determine whether the occurrence of other species may also play a role in shaping diel activity patterns of wildlife in this system.”

Reviewer #1, line-by-line comments:

• Abstract L32-33: I suggest caution in using the word “wildlife” when stating that the findings of the study imply a stronger influence of human features compared to direct human presence. For instance, the study focused on medium to large sized mammals (with the exception of the hare), thus the possibly differential impact on other small-sized mammals, such as small carnivores (e.g., mustelids) or micro-mammals, is not known, and such a generalization is potentially misleading. 

o We have added the phrase “medium to large-sized mammals” in place of “wildlife” (line 36). Additionally, we changed the word “wildlife” to “mammals” at line 245.

• L56: I think “human presence” is too general and its use should be limited to cases in which both human activities and human infrastructures are considered together. I suggest sticking to the definition of “direct human presence” used in Abstract L33 to properly maintain the two categories of human presence separated. The same is valid for all the other cases in the manuscript (e.g., line 106, 186, 206, 305, 314 etc.). 

o We understand this concern with our word choice, and have changed all instances of “human presence” to “direct human presence” to clarify our writing (lines 69, 132, 260, 272, 276, etc.)

• L70: human presence not only affects predator-prey interactions, but it could also affect temporal segregation among predators, thus possibly increasing intra-guild competition, see Wang et al. 2015, Biological Conservation for an example. 

o We thank you for pointing us towards this excellent example. We have included this citation, as well as the corresponding text “Risk effects may include [...] increased intra-guild competition...” to line 84.

• L93: I understand the emphasis on using continuous measurements of time, but I would suggest using “coarse” instead of “coarser”, as time of the day could be categorized in day, night and twilight, and in this case, with enough data, it could still be possible to properly detect crepuscular peaks in activity. 

o We have changed this instance of the word “coarser” to “coarse” as requested in line 111.

• L110: I do not like the use of “prevalent” in stating the first main hypothesis, as this term is never found again in the manuscript when dealing with the results and their implications for that specific hypothesis. I would suggest to avoid that specific word to remain more coherent with the rest of the manuscript. 

o We have removed the use of “prevalent” in this sentence, and restructured the sentence for clarity (lines 133-135). It now reads:

“We used these data in regression models to investigate whether human activity impacts wildlife nocturnality, testing the hypothesis that all wildlife species seek to minimize interactions with humans by exhibiting increased nocturnality in areas where human activity is greater (27).”

• L113-115: I suggest sticking to the singular form when presenting the species names for the first time, especially when associated with their scientific name, as the reference is to species in the widest sense of their definition. 

o We have changed the first mention of all species names to the singular form in lines 139-141.

• L130-131: This part of the manuscript links with my main critic, as the study area is not predator-free. For this reason, black-tailed deer might have shifted its diel activity pattern towards crepuscular periods to avoid humans, or predators, or even both. I pointed this out also to help answering my main critic.

o We recognize that this is a valid criticism of our predictions, but we have elected to keep this section written as-is, because this was our original prediction, and we believe that changing predictions upon viewing the results of an analysis is inappropriate. However, we hope that the changes we have made to the main criticism (i.e., addressing the omission of biotic variables in lines 345-365) will alleviate any concerns with this section.

• L142: I suggest to include an explication of the BC abbreviation for British Columbia, and to also include the country (Canada). I think it is important to give all the information a reader could need to properly understand the study (not all people around the world may know what or where BC is). 

o We have included the full wording of British Columbia, as well as the country (Canada) to line 58, at the first instance they are mentioned. We have also adjusted our Figure 1 to include inset maps of North America and British Columbia in order to further clarify to readers where this study took place.

• L184: Information about data extraction from camera trap images is missing.: software used for image managing (if any), and specific R packages or software used for event extraction (if any). 

o We have added information about data extraction from camera trap images to lines 249-257. Additionally, we have added information for all R packages where we previously overlooked this information (e.g., line 341).

• L200: The R package name should be accompanied by the package version. This is valid for all packages named through the manuscript. 

o We have added R package versions and citations for all packages named throughout the manuscript (e.g., lines 271, 333, 341, 371...)

• L208-212: I suggest sticking to the definition of weekly/monthly detection RATES correctly used in other parts of the manuscript (e.g., Table 1), as they are number of detections divided by the camera trapping effort. Furthermore, the definition given does not correspond to the one given in the TABLE S1 FILE caption, where it is stated that the variable refers to the detection rate for the week of the specific camera trap event. Please clarify this point and make it coherent throughout the manuscript. See also the associated comment for lines 700-702. 

o We agree that phrase “detection rates” more accurately describes the metric we have calculated, and we thank you for outlining instances of this where we were not consistent with this phrasing. We have changed all instances of this wording issue (lines 290, 292, 320, 321, 325, 459, etc.)

• L235: It is unclear to me why human infrastructure variables were evaluated in a buffer, whereas crown closure was considered only for the camera trapping site. I would like to ask for further details about this choice. 

o To address why we employed these differences in our methods of calculating these variables, we added the following rationale to lines 341-345: 

“We did not evaluate crown closure within a buffer (as done for trail density and road density), because this variable was meant to characterize light availability in the immediate vicinity of the CT, therefore controlling for a habitat characteristic which might impact the likelihood of wildlife using a specific area at night.”

We hope this reasoning addresses any concerns with this decision.

• L240-243: Information about eventual adaptation/warm-up phase and thinning are missing. Furthermore, I suggest to explicitly state the threshold used in evaluating convergence through the Gelman-Rubin Rhat. 

o We had already included information about the adaptation/warm-up phase (referred to as the “burn-in period” in lines 370-371), but we have added the thin rate and the specific threshold of the Gelman-Rubin R-hat which we used to lines 371-372:

“We constructed all models with flat priors (e.g., uniform distribution with bounds from -infinity to infinity), running them with 100,000 iterations across 4 chains (burn-in period = 5,000, thinning rate = 1) using the R package brms (68). We confirmed model convergence with the Gelman-Rubin statistic (R-hat < 1.1) (69) and by visually assessing trace plots.”

• L255: as for lines 208-212, consider that the ones within the day are detections, the ones in the week or month are detection rates. 

o As before, we have corrected this lack of clarity in our wording.

• L259 TAB1: “to THE anticipated” 

o We understand that this suggestion may be warranted in some cases, but for this particular sentence, the writing is grammatically correct as is. This is comparable to someone saying “due to unexpected circumstances” vs. “due to the unexpected circumstances”. Both are valid, but neither is a “correct” option. Therefore, we have not made the requested changes to the sentence, because the requested change is based on opinion, which will vary from reader to reader.

• L290 TAB: if I understood well, this should be table TWO, not table one. 

o The table at line 290 in the original manuscript (now line 438 in the revised manuscript) is already labeled “Table 2”. 

• L299 FIG2: “The x-axes […] and y-axes”, whether using the article or not, I suggest doing it coherently in this case. 

o We have added the word “the” before “y-axes” for consistency (line 448).

• L307; L309: again, those are detection rates. 

o As before, we have corrected this lack of clarity in our wording.

• L321; L377: I suggest using whereas instead of while in these case. 

o We have changed the word “while” to “whereas” in lines 473 and 544.

• L401: “utilize urban areas”, remove the A article. 

o Thank you for catching this mistake, we have corrected this (line 572).

• L453: “A continuous measure” 

o Thank you for also catching this mistake, we have corrected this (line 634).

• L478: “the temporal niche IN which both predators and prey occupy”, this statement is unclear to me; please provide further explanation or rephrase it. 

o We have rewritten the sentence in lines 657-660 as:

“In some cases, human activity may fundamentally alter trophic structures and natural predator-prey dynamics by restricting the temporal niches predators and prey may occupy, thereby preventing prey species from avoiding predators (36).”

We hope that the restructuring improves clarity.

• L700-702 TAB S1: It is unclear to me why you refer to “the average” detection rate for weeks and months in this case, whereas no reference to the average value of detection rates was made throughout the manuscript.

o As before, we have corrected all instances of this issue, and have ensured we consistently refer to these variables as “detection rates”.

Reviewer #2:

• Line 46: Instead of saying that ‘PAs are often thought to provide refuge..’ I would rather say that PAs should provide refuge from anthropogenic pressures, since we established PAs for that very reason, and if they do not offer any refuge from anthropogenic pressures then they are failing to fulfil their scope. I suggest you should deepen this point a little bit in this part of the Intro. 

o While we understand that many protected areas are indeed established with the intent of providing wildlife refuge from anthropogenic pressures, we would also like to acknowledge that this is not the sole reason that all protected areas are established. Therefore, we have expanded this section to include several other reasons that PAs might be established, and to emphasize that protected areas which are explicitly established for the purposes of both biodiversity conservation and recreation may be failing to meet the goal of reducing impacts on wildlife due to potential impacts of recreation on wildlife. The following is from lines 52-62:

“Protected areas (PAs), and specifically PAs created for the purpose of wildlife conservation, are often thought to provide refuge from anthropogenic pressures, lessening negative human impacts on wildlife (12,13). However, PAs may be established for a number of reasons including the maintenance of accessible outdoor spaces for the purposes of recreational activity to promote human health (14), the preservation of culturally important landmarks or features (15), or the conservation of unique ecosystems or species (e.g., Garry oak ecosystems in the Gulf Islands National Park Reserve in British Columbia (BC), Canada (16)). Many PAs established and maintained with a mandate to conserve wildlife are also intended to promote recreational activity. Yet, high human visitation rates—a growing feature of many of the world's PAs (17)—may hinder conservation efforts, displacing wildlife from regions intended to safeguard them (13,18).”

• Lines 64-67: Perhaps I misunderstood your point here, but what do you mean exactly when you say that different species may respond to human disturbance at different temporal scales? Why would fine-scale changes in human presence not affect species that avoid seasonal or monthly patterns in human disturbance? Is there any evidence of animals responding to e.g. increase in tourism in summer not responding to daily changes in human activity? 

o We understand there may be some confusion with this, and we have adjusted the wording in hopes of alleviating this misunderstanding at lines 77-80: 

“Wildlife responses to human disturbance may also vary with the temporal scale of human activity, with more behaviorally plastic species responding to fine-scale shifts in human presence (e.g., hourly or daily traffic (28,30)), while other species may exhibit patterns which change with weekly or seasonal trends in anthropogenic pressures (31–33).”

We have already provided several examples from the literature where species are shown to respond to different temporal scales of human activity. Therefore, we hope that by expanding on the idea that some species are more behaviorally plastic than others, we have clarified our intentions for this section.

• Line 73: you already used the verb underscore at line 56. 

o We have changed the instance of “underscore” in line 86 to “reinforce”.

• Lines 76-77: Is this ability really linked positively to body-mass? I would say is rather typical of generalist species, that can exploit disturbed areas by shifting to nocturnality, while more sensitive species usually avoid disturbed areas also spatially, or spatio-temporally and not only temporally. 

o We appreciate this point. After reviewing the sources we have cited in this sentence, we believe that these studies all mentioned that the study species in question are generally large-bodied. However, we overlooked that some of these studies also mention that the tendency towards increasing nocturnal behavior is less likely linked to body mass, and instead is more closely associated with a history of conflict with humans. Such is the case for most of the species in the studies we have cited in this sentence (e.g., cougars, coyotes, bobcats, and elephants), so we have made adjustments to lines 95-97 to reflect this alternative explanation:

“Some species—especially those with a history of conflict with humans—may be more adaptable (7,28,38), while others may predominantly exhibit nocturnal behavior regardless of external influences (e.g., snowshoe hares, Lepus americanus (39)).” 

• Line 92: Isn’t wildlife an uncountable noun? I think ‘wildlife is active’ is the correct form. 

o We appreciate this suggestion, but believe that the sentence is grammatically correct as written, since the word “wildlife” can also be the plural form of the singular word “wildlife”. We have therefore not made any changes to this line (now line 110)

• Line 95: Concerning human disturbance with social media data I think this research could be very useful here: Corradini, A., Randles, M., Pedrotti, L., van Loon, E., Passoni, G., Oberosler, V., ... & Cagnacci, F. (2021). Effects of cumulated outdoor activity on wildlife habitat use. Biological Conservation, 253, 108818. https://doi.org/10.1016/j. biocon.2020.108818 

o Thank you for providing us another great example of how social media data can be used to quantify human activity, especially in the context of wildlife research. We have added this recommended citation (line 114).

• Line 112: I find the term ‘human-related infrastructure’ odd, isn’t infrastructure intrinsically human related? 

o We acknowledge the phrasing of “human-related infrastructure” may be redundant. This was originally written to clarify that we observed infrastructure related to recreation in both landscapes, but also forest harvest and research-related infrastructure in the research forest. We have since changed this to say “land-use infrastructure” in the abstract (line 23) and the first instance in the main text (lines 137). All instances after this refer to these as “infrastructure”.

• Lines 141-147: When describing the study area you do not mention that British Columbia is part of Canada, nor in which part of Canada it is located. Readers outside north America might not know where British Columbia is, nor what the acronym BC stands for. 

o We have included the full wording of British Columbia, as well as the country (Canada) to lines 58 at the first instance they are mentioned. We have also adjusted our Figure 1 to include inset maps of North America and British Columbia in order to further clarify to readers where this study took place.

• Lines 153: I think you wanted to begin this sentence with ‘as a’ and not with ‘a’. 

o We believe that either option would have been grammatically correct, but this is a minor point. We have made this change (line 197).

• Line 163: I would erase the word ‘shaping’. 

o We have removed the word “shaping” (line 208).

• Lines 170-176: Here comes my main concern regarding your study: the sampling design is not based on a regular grid, thus cameras are set at very different distances one another. Sampling sites within Malcolm Knapp research forest are much closer compared to those within Golden Ears, and this may bias your results. Most probably sites in Malcolm Knapp will have a much higher level of spatial autocorrelation than those in Golden Ears. One way to overcome this would be to test if results of neighbouring sites are more correlated than those of sites that are more far apart (as for example done here: Kolowski, J. M., Oley, J., & McShea, W. J. (2021). High‐density camera trap grid reveals lack of consistency in detection and capture rates across space and time. Ecosphere, 12(2), e03350. https://doi.org/10.1002/ecs2.3350). 

o We appreciate the recommendation to make our work more statistically rigorous through this additional test. We have performed tests for spatial autocorrelation (Moran’s I), and have included the following statement about these tests in methods (lines 373-374):

“We also tested for spatial autocorrelation using Moran’s I tests for each species with R package spdep (70) (S1 Table).”

Additionally, we included the full results of these tests in supplementary materials (S1 Table). Furthermore, for one species (coyote), spatial autocorrelation was indeed detected. Therefore, we have added lines 491-493 stating:

“Moran’s I tests indicated a lack of spatial autocorrelation in all models with the exception of the coyote model (Moran’s I = 0.25, p = 0.04).”

We have also adjusted our discussion in lines 513-515 and 555-557 to reflect this new test:

“However, we also identified spatial autocorrelation in the coyote model, implying there may have been non-independence in coyote behavior across nearby sampling locations.” (lines 515-517)

“This potential shift in use between areas could potentially explain the spatial autocorrelation detected in the coyote model; we recommend additional modeling to more effectively understand the drivers of coyotes diel activities in this system.” (lines 557-559)

• Line 195: You now mention ‘shifts’ for the first time, and it is not very clear what you refer to here. More in general, I am not sure that shift is the right word to use for your variable, since it seems to indicate a deviation or change from a certain, known value, whereas you are indicating a temporal distance from noon, taken as a reference. 

o We thank you for pointing this out, and we agree with the points raised. We have removed this sentence to improve clarity (line 268).

• Lines 205-212: I think that here it is not well justified and explained why there would be a need to test human passage at these three temporal scales. I imagine that sites with high overall human passage also have high weekly and daily values. The only situation in which this does not hold would be if human frequentation was concentrated in a few days and scarce for the rest of the sampling period (is that the case for your study areas?), but this possibility can easily been tested beforehand. Also, if that was the case, could you please list how would mammalian species respond differently to disturbance occurring at these three temporal intervals, are there evidence and example from the literature? 

o We understand that this concern points out that daily, weekly, and monthly human detections are all collinear, and therefore, it may be unnecessary to test for differences in wildlife responses to each of these independently because responses might be similar regardless of the temporal scale. This is a fair point, but this analysis is simply another facet we thought was worth exploring, and retaining this analysis does not challenge the broader goals or conclusions of the research. Nevertheless, to address the reviewers’ concerns, we have adjusted lines 276-286 to reflect the proposed mechanism by which species might respond differently to different scales:

“We speculated that direct human presence might impact species’ behavior at different temporal scales, with some species exhibiting more plastic behavior (e.g., immediate responses to daily human use), while others might exhibit behavioral patterns that are aligned with longer periods of time (e.g., more consistent responses to anticipated seasonal changes in human use), and thus calculated three temporal scales of direct human presence”

Additionally, we had already included in our introduction some examples from the literature showing how wildlife may respond to human activity across different temporal scales, but we have added to this in lines 77-80 to further clarify this point:

“Wildlife responses to human disturbance may also vary with the temporal scale of human activity, with more behaviorally plastic species responding to fine-scale shifts in human presence (e.g., hourly or daily traffic (28,30)), while other species may exhibit patterns which change with weekly or seasonal trends in anthropogenic pressures (31–33).”

We hope that these changes address this concern adequately.

• Line 228: I think there should be a short explanation of what Bayes Factor factor is for readers that are not familiar with it. 

o We have added the following short explanation of Bayes Factor to lines 328-330:

“Bayes Factor represents a ratio of the likelihood of a hypothesis against the likelihood of a competing hypothesis (59)—in our case the likelihood of a direct human presence variable explaining more variation in the data than the null model.”

• Results section: Have you assessed if you get similar results using a different nocturnality index (as for example nocturnal/(diurnal+nocturnal) events ratio)? I wonder whether the choice of the response variable is affecting your analysis. 

o We have not assessed whether we get similar results using a different nocturnality index. We have stated in the discussion (lines 631-636) that: 

“[a previous study] used a coarse nocturnality measure of day/night rather than a continuous measure of temporal activity. This sort of analysis may interpret smaller-scale shifts in wildlife diel activity (e.g., species becoming more crepuscular) as full shifts towards nighttime activity.” 

Therefore, the point raised—that we might see different results with a categorical response variable—is valid, but we argue that the continuous response variable accounts for small changes toward crepuscular behavior which might be incorrectly interpreted while using categorical variables, and that this was a large reason why we elected to construct our models in this way.

• Line 285-287: This sentence is not well linked to what is written before and after. Why is this observation important, what does it entail? 

o We intended to use this result to illustrate that many species were detected during both daytime and nighttime hours, therefore offering some level of variation in our detection data. We have adjusted the sentence to reflect this rationale in lines 407-410:

“Nonetheless, all species were detected at least once within two decimal hours of solar noon, and at least once within one decimal hour of solar midnight (i.e., 12 decimal hours from solar noon), indicating some level of variation within all species’ diel activity patterns was present.”

• Line 364: This likewise here confounds me a little, you jump from effects of urban areas to seasonality too abruptly. 

o We understand that our use of the word “likewise” has created confusion in the interpretation of these sentences. Originally, this word was included to juxtapose previous findings regarding black bear seasonality, but there was no need for us to have split these findings into multiple sentences. We have now combined the sentence of concern with the previous sentence in order to clarify the wording of this section in lines 527-533:

“Little research has reported impacts of human activity on black bear diel activities, and of those which did, black bears were found to be more nocturnal at urban interface areas (50), to adjust their diel activities in response to seasonality (76), and/or to maintain mostly crepuscular activity patterns regardless of external influences (77).”

• Line 401: erase the ‘a’ from ‘a urban spaces’. 

o We thank you for catching this mistake. We have removed the word “a” (line 572).

• Lines 413-415: Did you explicitly test whether roads and trails were more used in Golden Ears than Malcolm Knapp? It would be useful to report numerical output of such test here.

o We recognize that the differences between traffic in each landscape had not previously been shown empirically, and we thank you for cluing us into this oversight. We have since included the following information regarding the average number of people detected in each area (total human detections per area divided by the number of cameras in each area) in lines 585-587:

“Roads in Malcolm Knapp are not as heavily used as roads or recreation trails in Golden Ears (Golden Ears averaged 3728 humans per camera station over the entire study period while Malcolm Knapp only averaged 272).”

---

## [Decision Letter · Decision Letter 1]

10 Apr 2023

PONE-D-22-31729R1Human presence and infrastructure impact wildlife nocturnality differently across an assemblage of mammalian speciesPLOS ONE

Dear Dr. Procko,

Thank you for submitting your manuscript to PLOS ONE. After careful consideration, we feel that it has merit but does not fully meet PLOS ONE’s publication criteria as it currently stands. Therefore, we invite you to submit a revised version of the manuscript that addresses the points raised during the review process. I have obtained two reviewers of your resubmission and both concur that you have done a good and diligent work at revising the submission. Reviewers raise a few remaining issues (please consider also the attached review from Reviewer 1) that I recommend addressing before the manuscript can be accepted for publication.

We look forward to receiving your revised manuscript.

Kind regards,

Francesco Rovero, Ph.D.

Academic Editor

PLOS ONE

Journal Requirements:

Additional Editor Comments:

I have obtained two reviewers of your resubmission and both concur that you have done a good and diligent work at revising the submission. Reviewers raise a few remaining issues that I recommend addressing before the manuscript can be accepted for publication.

Reviewers' comments:

Reviewer's Responses to Questions

**Comments to the Author**

1. If the authors have adequately addressed your comments raised in a previous round of review and you feel that this manuscript is now acceptable for publication, you may indicate that here to bypass the “Comments to the Author” section, enter your conflict of interest statement in the “Confidential to Editor” section, and submit your "Accept" recommendation.

Reviewer #1: (No Response)

Reviewer #2: All comments have been addressed

2. Is the manuscript technically sound, and do the data support the conclusions?

Reviewer #1: Yes

Reviewer #2: Yes

3. Has the statistical analysis been performed appropriately and rigorously? 

Reviewer #1: Yes

Reviewer #2: Yes

4. Have the authors made all data underlying the findings in their manuscript fully available?

Reviewer #1: Yes

Reviewer #2: (No Response)

5. Is the manuscript presented in an intelligible fashion and written in standard English?

Reviewer #1: Yes

Reviewer #2: Yes

6. Review Comments to the Author

Reviewer #1: Overall, the manuscript is well written, and I think it meets the rigorous standards for publication in the journal. The authors fairly addressed the requests I had on the first version of the manuscript, resolving the main criticism I moved to it. However, I still have a minor concern about the manuscript, and I think a some corrections are required.

L81: I appreciated your inclusion of the work from Wang et al., which explores another possible effect of human activity on wildlife; however, I would suggest to rephrase this part in something like:

“increased temporal overlap between predator and prey species (causing greater predation risk (36)), and between competing predators (37), or effective habitat loss […]”

since Wang et al. found a reduction in temporal segregation among coexisting predators, but they did not directly test whether this resulted in increased competition or not (as they clearly state in their discussion).

L216: Please provide the name and version of the software used to manually process the camera trap images.

L361 TABLE: I have to insist on this point. In the new version of the manuscript I still find a Table 1 at line 258, describing the “Predictor variables considered[…]”, which is correctly reported as Table 1 in the text. Then, on line 361 I find a second table, still named Table 1: “Summary of detections […]”, which is (correctly) reported within text as Table 2. Please clarify this.

L345-348: This is the only (minor) concern I have about the manuscript. I think presenting the activity patterns of coyotes together with that of cougars and deer is a bit misleading. It is true that coyote activity has two “peaks” around dusk and down, but I would suggest caution in defining its activity as bimodal, as it never really drops during the night, conversely to what happens to cougar and especially deer activity. I would suggest to clarify that, also considering that based on Table 2 coyote is the third more nocturnal species (after hare and bobcat).

Reviewer #2: I thank the Authors for carefully answering and considering all of my comments of the previous round. I think that after considering the few more comments I have given in this second round of review the manuscript will be ready to be published.

7. PLOS authors have the option to publish the peer review history of their article (what does this mean?). If published, this will include your full peer review and any attached files.

Reviewer #1: No

Reviewer #2: No

---

## [Author Response · Author response to Decision Letter 1]

18 Apr 2023

Dear Dr. Rovero,

Thank you again for facilitating a timely review of our manuscript. We also thank the reviewers for their time spent reviewing our manuscript and the thoughtful feedback they provided. We have revised the manuscript according to the requested changes, and we hope that this improved manuscript is now suitable for publication in PLOS ONE. Below you will find our point-by-point responses to the reviewer’s recommendations (line numbers of reviewer comments refer to the original manuscript, but line numbers in our responses refer to the revised manuscript). We look forward to hearing from you and thank you again for your consideration. 

Reviewer #1, general comments:

• Overall, the manuscript is well written, and I think it meets the rigorous standards for publication in the journal. The authors fairly addressed the requests I had on the first version of the manuscript, resolving the main criticism I moved to it. However, I still have a minor concern about the manuscript, and I think a some corrections are required.

o We thank you for these comments. We have made revisions as requested, and hope that our manuscript now adequately addresses the issues raised.

Reviewer #1, line-by-line comments:

• L81: I appreciated your inclusion of the work from Wang et al., which explores another possible effect of human activity on wildlife; however, I would suggest to rephrase this part in something like:

“increased temporal overlap between predator and prey species (causing greater predation risk (36)), and between competing predators (37), or effective habitat loss […]”

since Wang et al. found a reduction in temporal segregation among coexisting predators, but they did not directly test whether this resulted in increased competition or not (as they clearly state in their discussion).

o We have changed lines 81-82 to reflect this request:

“...increased temporal overlap between predator and prey species (causing greater predation risk (36)), increased temporal overlap between competing predators (37), or effective habitat loss...” 

• L216: Please provide the name and version of the software used to manually process the camera trap images.

o We have added the name and version of the private software to lines 217-218:

“...private cloud-based photo management software (WildCo CamTrap System v 3.0) (54).”

• L361 TABLE: I have to insist on this point. In the new version of the manuscript I still find a Table 1 at line 258, describing the “Predictor variables considered[…]”, which is correctly reported as Table 1 in the text. Then, on line 361 I find a second table, still named Table 1: “Summary of detections […]”, which is (correctly) reported within text as Table 2. Please clarify this.

o We apologize for misunderstanding your original request. Line 363 now reads “Table 2”

• L345-348: This is the only (minor) concern I have about the manuscript. I think presenting the activity patterns of coyotes together with that of cougars and deer is a bit misleading. It is true that coyote activity has two “peaks” around dusk and down, but I would suggest caution in defining its activity as bimodal, as it never really drops during the night, conversely to what happens to cougar and especially deer activity. I would suggest to clarify that, also considering that based on Table 2 coyote is the third more nocturnal species (after hare and bobcat).

o We have changed lines 346-354 to reflect this request:

“Cougars and black-tailed deer maintained diel activity curves with mostly bimodal distributions, peaking at or near dawn and dusk (cougar activity peaked around 6:00 am and 9:00 pm, while black-tailed deer activity peaked at approximately 6:00 am and 6:00 pm; Fig. 2). Black bear activity was greatest from approximately 6:00 am to 9:00 pm, with activity steadily increasing from around 2:00 pm to 9:00 pm where it peaked and subsequently dropped (Fig. 2). Snowshoe hares, coyotes, and bobcats all showed nocturnal activity patterns, with snowshoe hare activity being greatest from approximately 10:00 pm to 4:00 am, coyote activity peaking around 6:00 am and 9:00 pm (and remaining active during nighttime hours), and bobcat activity peaking around 2:00 am (Fig. 2).”

Reviewer #2, general comments:

• I thank the Authors for carefully answering and considering all of my comments of the previous round. I think that after considering the few more comments I have given in this second round of review the manuscript will be ready to be published.

o It does not appear that there were any additional comments from this reviewer. Nevertheless, we appreciate the reviewer taking the time to read our manuscript again, and hope that our latest revisions adequately address any remaining concerns.

---

## [Decision Letter · Decision Letter 2]

27 Apr 2023

PONE-D-22-31729R2Human presence and infrastructure impact wildlife nocturnality differently across an assemblage of mammalian speciesPLOS ONE

Dear Dr. Procko,

Thank you for submitting your manuscript to PLOS ONE. After careful consideration, we feel that it has merit but does not fully meet PLOS ONE’s publication criteria as it currently stands. Therefore, we invite you to submit a revised version of the manuscript that addresses the points raised during the review process. Please find attached in a pdf the additional comments from Reviewer 2 which were not apparently attached in the earlier communication. 

We look forward to receiving your revised manuscript.

Kind regards,

Francesco Rovero, Ph.D.

Academic Editor

PLOS ONE

Journal Requirements:

Additional Editor Comments:

Please find some additional comments from Reviewer 2 in the pdf attached which were not apparently attached to the previous communication. I look forward to receiving a further revision that addresses these comments.

Reviewers' comments:

Reviewer's Responses to Questions

**Comments to the Author**

1. If the authors have adequately addressed your comments raised in a previous round of review and you feel that this manuscript is now acceptable for publication, you may indicate that here to bypass the “Comments to the Author” section, enter your conflict of interest statement in the “Confidential to Editor” section, and submit your "Accept" recommendation.

Reviewer #2: All comments have been addressed

2. Is the manuscript technically sound, and do the data support the conclusions?

Reviewer #2: Yes

3. Has the statistical analysis been performed appropriately and rigorously? 

Reviewer #2: Yes

4. Have the authors made all data underlying the findings in their manuscript fully available?

Reviewer #2: Yes

5. Is the manuscript presented in an intelligible fashion and written in standard English?

Reviewer #2: Yes

6. Review Comments to the Author

Reviewer #2: (No Response)

7. PLOS authors have the option to publish the peer review history of their article (what does this mean?). If published, this will include your full peer review and any attached files.

Reviewer #2: No

---

## [Author Response · Author response to Decision Letter 2]

8 May 2023

Dear Dr. Rovero,

Thank you again for facilitating a timely review of our manuscript. We also thank the reviewers for their time spent reviewing our manuscript and the thoughtful feedback they provided. We have revised the manuscript according to the requested changes, and we hope that this improved manuscript is now suitable for publication in PLOS ONE. Below you will find our point-by-point responses to the reviewer’s recommendations (line numbers of reviewer comments refer to the original manuscript, but line numbers in our responses refer to the revised manuscript). We look forward to hearing from you and thank you again for your consideration. 

Reviewer #1, general comments:

• Overall, the manuscript is well written, and I think it meets the rigorous standards for publication in the journal. The authors fairly addressed the requests I had on the first version of the manuscript, resolving the main criticism I moved to it. However, I still have a minor concern about the manuscript, and I think a some corrections are required.

o We thank you for these comments. We have made revisions as requested, and hope that our manuscript now adequately addresses the issues raised.

Reviewer #1, line-by-line comments:

• L81: I appreciated your inclusion of the work from Wang et al., which explores another possible effect of human activity on wildlife; however, I would suggest to rephrase this part in something like:

“increased temporal overlap between predator and prey species (causing greater predation risk (36)), and between competing predators (37), or effective habitat loss […]”

since Wang et al. found a reduction in temporal segregation among coexisting predators, but they did not directly test whether this resulted in increased competition or not (as they clearly state in their discussion).

o We have changed lines 81-82 to reflect this request:

“...increased temporal overlap between predator and prey species (causing greater predation risk (36)), increased temporal overlap between competing predators (37), or effective habitat loss...” 

• L216: Please provide the name and version of the software used to manually process the camera trap images.

o We have added the name and version of the private software to lines 217-218:

“...private cloud-based photo management software (WildCo CamTrap System v 3.0) (54).”

• L361 TABLE: I have to insist on this point. In the new version of the manuscript I still find a Table 1 at line 258, describing the “Predictor variables considered[…]”, which is correctly reported as Table 1 in the text. Then, on line 361 I find a second table, still named Table 1: “Summary of detections […]”, which is (correctly) reported within text as Table 2. Please clarify this.

o We apologize—we misunderstood your original request. Line 363 now reads “Table 2”

• L345-348: This is the only (minor) concern I have about the manuscript. I think presenting the activity patterns of coyotes together with that of cougars and deer is a bit misleading. It is true that coyote activity has two “peaks” around dusk and down, but I would suggest caution in defining its activity as bimodal, as it never really drops during the night, conversely to what happens to cougar and especially deer activity. I would suggest to clarify that, also considering that based on Table 2 coyote is the third more nocturnal species (after hare and bobcat).

o We have changed lines 346-354 to reflect this request:

“Cougars and black-tailed deer maintained diel activity curves with mostly bimodal distributions, peaking at or near dawn and dusk (cougar activity peaked around 6:00 am and 9:00 pm, while black-tailed deer activity peaked at approximately 6:00 am and 6:00 pm; Fig. 2). Black bear activity was greatest from approximately 6:00 am to 9:00 pm, with activity steadily increasing from around 2:00 pm to 9:00 pm where it peaked and subsequently dropped (Fig. 2). Snowshoe hares, coyotes, and bobcats all showed nocturnal activity patterns, with snowshoe hare activity being greatest from approximately 10:00 pm to 4:00 am, coyote activity peaking around 6:00 am and 9:00 pm (and remaining active during nighttime hours), and bobcat activity peaking around 2:00 am (Fig. 2).”

Reviewer #2, line-by-line comments:

• Discussion, Lines 453-455: I think this is an interesting finding and I would maybe discuss it more broadly. Most of the species you targeted show on average a shift towards nocturnal behavior in presence of humans, even if BCI are not strictly ‘significant’ I still think it is an interesting finding. 

o We agree that this is an interesting finding that deserves more discussion beyond what is briefly stated in lines 453-455. However, we already addressed this in other areas of the manuscript. In lines 507-530, we further discuss the trends of all carnivores being predominantly more nocturnal, despite a lack of “significance”. Specifically, in lines 510-516, we discuss how human activity is predictably diurnal in this study area, which may allow carnivores to adequately segregate by becoming only slightly more nocturnal (e.g., becoming more crepuscular). We then mention how small shifts towards increased crepuscular activity might lead to smaller effect sizes in lines 516-518, which could help to explain the lack of “significance”. Furthermore, in lines 524-530 we discuss how individual behavioral variation may have produced greater uncertainty (wider CIs) in our estimates and made it more difficult to make broader claims about the overall “population-level” impacts on these diverse individuals. We aimed to keep these speculative discussion points to a minimum, but we hope that the ideas raised in this section adequately discuss the broader implications of this finding.

• Lines 479-481: Are these species really naturally nocturnal? Is there evidence from non-human-dominated landscapes that these species still maintain nocturnality? Even species classically considered strictly crepuscular or nocturnal have been reported to be more diurnal where human presence is low (see for example: Kamler, J. F., Jędrzejewska, B., & Jędrzejewski, W. (2007). Activity patterns of red deer in Białowieża National Park, Poland. Journal of mammalogy, 88(2), 508-514.)

o This is a very interesting point—thank you for bringing this up. To address this, we have added lines 455-460:

“...additional research would be needed to confirm whether these species are indeed nocturnal in areas without human presence. Some species which are typically thought of as nocturnal or crepuscular may exhibit diurnal behavior in areas where humans or predators are excluded (51,78), so there is also a need for additional monitoring in areas without humans to understand the full range of wildlife diel activities both with and without human influences.”

• Line 587: I think this work is highly relevant here: Salvatori, M., Oberosler, V., Rinaldi, M., Franceschini, A., Truschi, S., Pedrini, P., & Rovero, F. (2023). Crowded mountains: Long-term effects of human outdoor recreation on a community of wild mammals monitored with systematic camera trapping. Ambio, 1-13. https://doi.org/10.1007/s13280-022-01825-w Indeed, increased nocturnality and spatial avoidance of humans might not directly translate into decreased fitness for mammals, at least in the temporal interval considered in this study (10 years). So spatio-temporal changes in behavior to decrease the probability to encounter humans could actually be effective strategies to cope with recreation. However I agree with you that some limitations to human accessibility in PAs are desirable, especially considering the increasing trends in outdoor recreation across the globe.

o We thank you for including this new paper. We have incorporated it into our final discussion point at lines 570-572:

“... long-term camera trapping studies have also shown increasing occupancy of high human use areas by of a number of wildlife species when they are able to adequately segregate via increased nocturnality (88).”

---

## [Editor Report · Decision Letter 3]

10 May 2023

Human presence and infrastructure impact wildlife nocturnality differently across an assemblage of mammalian species

PONE-D-22-31729R3

Dear Dr. Procko,

We’re pleased to inform you that your manuscript has been judged scientifically suitable for publication and will be formally accepted for publication once it meets all outstanding technical requirements.

Kind regards,

Francesco Rovero, Ph.D.

Academic Editor

PLOS ONE

---

## [Editor Report · Acceptance letter]

17 May 2023

PONE-D-22-31729R3 

Human presence and infrastructure impact wildlife nocturnality differently across an assemblage of mammalian species 

Dear Dr. Procko:

I'm pleased to inform you that your manuscript has been deemed suitable for publication in PLOS ONE. Congratulations! Your manuscript is now with our production department. 

Kind regards, 

on behalf of

Dr. Francesco Rovero 

Academic Editor

PLOS ONE